# Deep Generalized Prediction Set Classifier and Its Theoretical Guarantees

**Zhou Wang**  *zwang198@binghamton.edu*
*Department of Mathematics and Statistics*
*Binghamton University, the State University of New York*

**Xingye Qiao**  *xqiao@binghamton.edu*
*Department of Mathematics and Statistics*
*Binghamton University, the State University of New York*

**Reviewed on OpenReview:** *https://openreview.net/forum?id=H7gLN5nqVF*

## Abstract

A standard classification rule returns a single-valued prediction for any observation without a confidence guarantee, which may result in severe consequences in many critical applications when the uncertainty is high. In contrast, set-valued classification is a new paradigm to handle the uncertainty in classification by reporting a set of plausible labels to observations in highly ambiguous regions. In this article, we propose the Deep Generalized Prediction Set (DeepGPS) method, a network-based set-valued classifier induced by acceptance region learning. DeepGPS is capable of identifying ambiguous observations and detecting out-of-distribution (OOD) observations. It is the first set-valued classification of this kind with a theoretical guarantee and scalable to large datasets. Our nontrivial proof shows that the risk of DeepGPS, defined as the expected size of the prediction set, attains the optimality within a neural network hypothesis class while simultaneously achieving the user-prescribed class-specific accuracy. Additionally, by using a weighted loss, DeepGPS returns tighter acceptance regions, leading to informative predictions and improved OOD detection performance. Empirically, our method outperforms the baselines on several benchmark datasets.

## 1 Introduction

A standard classification method assigns only a single class label to each test observation, regardless of its confidence toward this prediction. However, this approach might be problematic in critical domains where even a single incorrect decision can lead to disastrous consequences, such as in medical imaging-based diagnosis, autonomous driving systems, and military operations. Additionally, such a paradigm falls short in effectively controlling class-specific outcomes, especially in scenarios of imbalanced data. For instance, in medical diagnosis, it may incorrectly prioritize majority groups that do not need immediate attention while overlooking minority groups with certain diseases that demand urgent attention. This skewed prioritization results in delayed treatments, and ultimately, compromised patient outcomes. Lastly, conventional classification approaches often assume there is no distribution shift between the training and the test data, which is not the case in the open world. If a new class emerges, e.g., a new variant of a virus in the pandemic, it is imperative to detect out-of-distribution points. Therefore, there is a crucial need for novel methods that can simultaneously address these issues and deliver a reliable and risk-controllable decision in high-stake fields.

To mitigate the risks associated with conventional single-valued predictions, classifiers can first report multiple plausible labels for ambiguous observations in overlapped class regions (see Fig. 1). This approach allows for human intervention or secondary classification with additional features, ultimately reducing the risk of imprudent predictions. This has motivated the development of set-valued classification methods, which can be implemented in various ways. Classification with Reject Option (CRO) (Herbei & Wegkamp, 2006; Bartlett & Wegkamp, 2008; Zhang et al., 2018; Charoenphakdee et al., 2021) interprets a rejection of a difficult

observation as assigning all class labels to it, and trains the classifier by incorporating the rejection cost in the objective function. However, this method does not offer controlled misclassification rates for classes of interest, and it pertains to the closed-world setting.

To secure a trustworthy misclassification rate, Conformal Prediction (CP) (Vovk et al., 2005; Lei et al., 2013; 2015), a popular model-free framework, is developed in the machine learning community. With its theoretical guarantee on the prediction error rate, CP provides a safety solution in critical applications by generating prediction sets that encompass multiple plausible labels. Alternatively, Classification with Confidence (Lei, 2014; Sadinle et al., 2019; Wang & Qiao, 2018; 2022; Lin et al.,

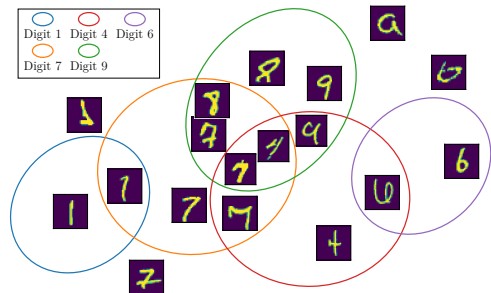

Figure 1: Illustration for MNIST class regions. Ambiguous points inside overlapped circles, OODs outside all circles.

2022) further optimizes the prediction set from a different perspective, aiming to yield the smallest prediction set while controlling the prescribed class-specific error rate. However, all these approaches focusing on the closed world are limited to generalizing their capability of out-of-distribution detection in the open world, as they are not tailored to this task.

In practice, data distribution may evolve and observations outside of existing classes in the training data may appear in the test data. To address this challenge, out-of-distribution (OOD) detection or open-set recognition (OSR) techniques (Bendale & Boult, 2015; Vaze et al., 2022; Kim et al., 2023) have been developed to detect the OOD class in addition to classification. Note that their single-valued decision-making rules still suffer the aforementioned restrictions. To overcome these limitations, researchers have proposed set-valued classifiers to detect OOD samples with controlled misclassification rates on each existing class, namely, Cautious Deep Learning (CDL) (Hechtlinger et al., 2018), Balanced and Conformal Optimized Prediction Set (BCOPS) (Guan & Tibshirani, 2022), and Generalized Prediction Set (GPS) (Wang & Qiao, 2023). However, all three methods are trained in a decoupled way, which may discard the underlying dependence among the learned acceptance regions across all classes. Additionally, besides GPS relies on computationally intensive quadratic programming, all three are shallow methods, making it challenging to scale the above methods to large datasets. Moreover, as per empirical performances, the GPS results in conservative acceptance regions due to the use of hinge loss, leading to sub-optimal finite-sample performances on both prediction set size and OOD detection; the CDL and BCOPS lack optimality on the empirical prediction set size, partially due to the fact that the prediction set size is not explicitly minimized.

In light of the limitations of current single-valued and set-valued prediction approaches, we propose an end-to-end Deep Generalized Prediction Set (DeepGPS) classifier jointly learning acceptance regions with several contributions. First, it generalizes and scales the set-valued classification to OOD detection by using a hypothesis class induced by neural networks and a kernel. To avoid relying on the expensive memory and quadratic programming for kernel machines, we add to the neural network a layer that approximates the kernel by using Random Fourier Features. Second, we provide nontrivial proof that shows the true accuracy of our classifier is bounded as the prescribed value, and that the expected prediction set size converges to the minimum within the hypothesis class. Third, we use an adaptive weighted loss to address the issue of GPS where the surrogate loss potentially produces larger acceptance regions. The weighted loss yields tighter acceptance regions, improving classification efficiency (defer to Section 2.1) and OOD detection performance.

## 2   Related Work

In this section, we introduce the notion of acceptance regions and terminologies in set-valued classification, and briefly discuss some related works. Note that there is a distinction between set-valued classification and multi-label classification (Zhang & Zhou, 2007). In set-valued classification, an observation has only one true label, whereas, in multi-label classification, there are multiple ground truths. Throughout the article, we use the notation $[K]$ to denote $1, \ldots, K$.

### 2.1 Set-valued Classification

Consider the multicategory classification with input space $\mathcal{X} = \mathbb{R}^p$ and label space $\mathcal{Y} = \{1, \ldots, K\}$. Given a rule, the set of observations classified as class $k$, $\mathcal{C}_k \subset \mathcal{X}$, is called the acceptance region for class $k$; all $K$ acceptance regions collectively induce a set-valued classifier $\phi : \mathcal{X} \to 2^{\mathcal{Y}}$ with $\phi(\boldsymbol{x}) := \{k \in [K] : \boldsymbol{x} \in \mathcal{C}_k\}$. Intuitively, there is a trade-off between the misclassification rate $\mathbb{P}(Y \notin \phi(\boldsymbol{X}))$, i.e., the probability of a set not containing the true class label, and the expected size of prediction set $|\phi(\boldsymbol{X})| := \sum_{k=1}^{K} \mathbb{1}\{\boldsymbol{X} \in \mathcal{C}_k\}$. A lower misclassification rate may require a larger prediction set. While two set-valued classifiers may have the same misclassification rate, the one with a smaller prediction size is more efficient/informative (the more efficient, the better the prediction set). We interchangeably use efficiency (see the definition in Appendix B) and prediction set size to denote the informativeness of a prediction set.

In the method of Classification with Reject Option (CRO) (Herbei & Wegkamp, 2006; Bartlett & Wegkamp, 2008; Charoenphakdee et al., 2021), the Bayes optimal rule under the 0-$d$-1 loss (where $d \in [0, 1 - 1/K]$ is the rejection cost and 1 is the misclassification cost) assigns an ambiguous observation $\boldsymbol{x}$ all labels if $\max_k \mathbb{P}(Y = k \mid \boldsymbol{x}) \leq 1 - d$, but a single label corresponding that with the largest probability score otherwise. Zhang et al. (2018) extended CRO with an additional refine option, which can output a smaller prediction set with size $1 < |\phi(\boldsymbol{x})| < K$ for those less difficult observations. CRO controls how many observations are rejected by changing the rejection cost $d$ (a smaller $d$ leads to more rejections). While this can improve the accuracy for those observations not rejected, there is no direct control over the classification accuracy.

In contrast, the Conformal Prediction set (Vovk et al., 2005; Lei et al., 2013; 2015) theoretically guarantees the accuracy $\mathbb{P}(Y \in \phi(\boldsymbol{X}))$. However, Conformal Prediction does not aim to maximize the efficiency of the classifier, i.e., there is no guidance on how to make the prediction set as small, and hence as informative as possible. In particular, a classifier with prediction set size $|\phi(\boldsymbol{x})| \equiv K$ for all $\boldsymbol{x}$ is useless even though its accuracy is 100%, while the single-valued prediction ($|\phi(\boldsymbol{x})| \equiv 1$ for all $\boldsymbol{x}$) might not be accurate albeit its 100% efficiency. To take into account the efficiency and accuracy simultaneously, Lei (2014), Sadinle et al. (2019), and Wang & Qiao (2018; 2022) minimize the expected prediction set size $\mathbb{E}[|\phi(\boldsymbol{X})|]$ while controlling the class-specific misclassification rate $\mathbb{P}(Y \notin \phi(\boldsymbol{X}) \mid Y = k) \leq \gamma$ specified by the user. In duality, Denis & Hebiri (2017; 2020) proposed to maximize the accuracy subject to a budget of prediction set size.

### 2.2 Out-of-distribution Detection and Selective Classification

Anomaly detectors aim to identify anomaly points not from the existing/normal class. One-Class Support Vector Machine (OCSVM) (Schölkopf et al., 2000) and Support Vector Data Description (SVDD) (Tax & Duin, 2004) are shallow detectors whose detection performance is improved with a kernel. To obtain better feature representations for large and complex data, Ruff et al. (2018) extended SVDD to Deep Support Vector Data Description by substituting neural networks for kernels.

Beyond the task of detecting/rejecting anomaly points not belonging to any of the normal classes, Out-of-distribution (OOD) detection and Open-set recognition (OSR) (Yang et al., 2021; Bendale & Boult, 2015; 2016) additionally conduct standard classification for normal observations. In contrast, Selective Classification (El-Yaniv et al., 2010; Geifman & El-Yaniv, 2017; Granese et al., 2021) centers on rejecting difficult normal observations besides single-valued classification. Different from CRO, it does not equate this type of rejection with assigning all labels to an observation. By allowing rejecting OOD and difficult normal observations, Xia & Bouganis (2022); Kim et al. (2023); Cen et al. (2023); Zhu et al. (2023) studied Selective Classification with OOD Detection (SCOD). However, this unified rejection mixes up the OOD and normal observations, which may obstruct the downstream task since one may impose different strategies on different types of rejections.

All the aforementioned classification methods with OOD detection still are attributed to the camp of single-valued classification, and hence suffer some issues highlighted in Section 1. In contrast, CDL, BCOPS, and GPS are set-valued approaches: they learn acceptance regions that collectively induce a prediction set to cover the true label with an advertised accuracy for normal points and reject potential OOD points. In particular, the prediction set $\phi(\boldsymbol{x})$ comprises all the classes $k \in [K]$ whose acceptance region $\mathcal{C}_k$ contains $\boldsymbol{x}$; when $\phi(\boldsymbol{x})$ is empty, $\boldsymbol{x}$ is marked as an OOD point.

## 3 Methodology

In this section, we formulate the optimization problem of DeepGPS. Suppose that a distribution $\mathcal{P}$ exclusively consists of $K$ (known) normal classes, while a target distribution $\mathcal{Q}$ may contain an (unknown) OOD component. To facilitate our analysis, we introduce two key assumptions.

**Assumption 1.** $p_{\mathcal{P}}(\boldsymbol{x} \mid Y = k) = p_{\mathcal{Q}}(\boldsymbol{x} \mid Y = k)$ *holds true for all normal classes* $k \in [K]$.

**Assumption 2.** *We have access to labeled data from* $\mathcal{P}$ *and unlabeled data from* $\mathcal{Q}$.

The equal class-conditional density in Assumption 1 is commonly employed to characterize "semantic shift" in the OOD detection literature (Yang et al., 2021; Garg et al., 2022) due to the emergence of a novel class. For instance, in the sentiment analysis of product reviews, the established sentiments such as positive or negative exhibit consistent linguistic patterns between past and current data, including the choice of words and sentence structures. However, when a review expresses entirely new sentiments, it represents an instance of OOD data. Assumption 1 is often accompanied by the mild Assumption 2 dealing with the utilization of unlabeled data (Du Plessis et al., 2015; Guan & Tibshirani, 2022; Garg et al., 2022; Katz-Samuels et al., 2022).

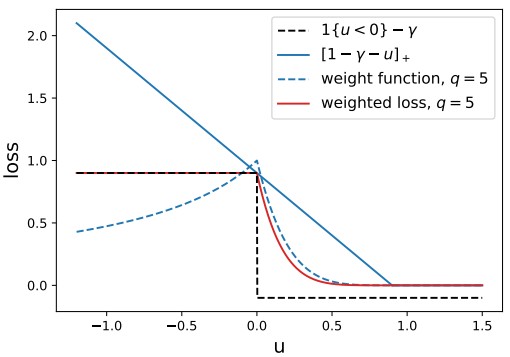

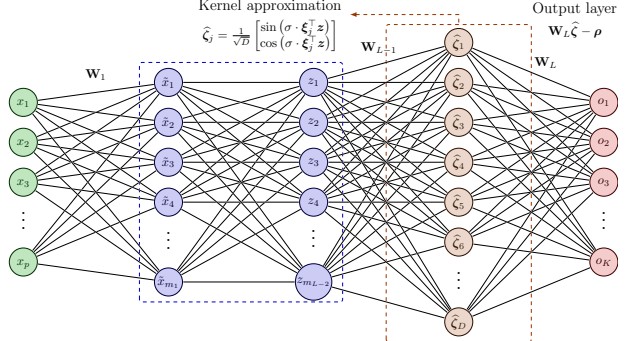

Figure 2: Functions with $\gamma = 0.1$. The blue bent solid line denotes loss function $\ell_{2,\gamma}(\cdot)$. The dashed blue curve denotes weight function $\omega_{k,j}$ in (3). The red curve represents the weighted loss in (4).

Figure 3: Network architecture. Each neuron in the penultimate layer, i.e., the kernel approximation layer, outputs a 2-d vector $\widehat{\boldsymbol{\zeta}}_j, j \in [D]$. The weight matrix $\mathbf{W}_{L-1}$ consists of random samples $\boldsymbol{\xi}_j \sim \mathcal{N}(\mathbf{0}, \mathbf{I}_{m_{L-2}})$.

**Remark 1.** *Several promising strategies are rooted in the existing literature to relax Assumption 1, hence accommodating tractable differences in class-conditional densities between training and test sets. Specifically, (1) Tachet des Combes et al. (2020) assumes that the identical class-conditional density is maintained within a feature space, facilitated by a representation mapping* $\tilde{g}$ *with* $p_{\mathcal{P}}(\tilde{g}(\boldsymbol{X}) \mid Y = k) = p_{\mathcal{Q}}(\tilde{g}(\boldsymbol{X}) \mid Y = k)$. *This approach, however, necessitates an additional assumption regarding the partitioning within its feature space. (2) The work of Wu et al. (2019); Kumar et al. (2020) delves into a relaxed label shift scenario, assuming a small divergence between class-conditional distributions. Nevertheless, determining the extent of this divergence in practice poses significant challenges, as highlighted by Garg et al. (2023). (3) Investigations by Zhang et al. (2013); Gong et al. (2016) into the location-scale generalized target shift (LS-GeTarS) assume the existence of an affine transformation affecting each dimension of* $\boldsymbol{X}$ *given* $Y$ *between the source and target domains. Employing the kernel embedding method (Fukumizu et al., 2007; Sriperumbudur et al., 2011) enables the alignment of the distribution between labeled data from the source domain and unlabeled data from the target domain. In the context of location-scale generalized target shift for normal classes, DeepGPS could be applied to the test data and the transformed training data returned from the LS-GeTarS method.*

*Regarding the relaxation of Assumption 2 involving additional unlabeled data, DeepGPS still works with the generated auxiliary data (Hendrycks et al., 2019; Du et al., 2022) even if one cannot access the test data. On the other hand, without accessing all data points, DeepGPS's application extends seamlessly to online learning environments, where the model is updated by optimizing objective function Eq. (1) with stochastic gradient descent on each batch of data.*

Let $(\boldsymbol{X}, Y) \in \mathcal{X} \times \mathcal{Y}$ come from the distribution $\mathcal{Q}$, where $\mathcal{X} = \mathbb{R}^p$ and $\mathcal{Y} = \{\text{OOD}, 1, 2, \ldots, K\}$. Let $\boldsymbol{f}(\boldsymbol{x}) = (f_1(\boldsymbol{x}), \ldots, f_K(\boldsymbol{x}))^\top$ be a vector of decision functions for normal classes, which induce the acceptance regions through $\mathcal{C}_k := \{\boldsymbol{x} : f_k(\boldsymbol{x}) \geq 0\}, k \in [K]$. Then the set-valued classifier is defined as $\phi : \mathcal{X} \to 2^{\mathcal{Y}}$ with a prediction $\phi(\boldsymbol{x}) := \{k \in [K] : \boldsymbol{x} \in \mathcal{C}_k\} = \{k \in [K] : f_k(\boldsymbol{x}) \geq 0\}$ for a query $\boldsymbol{x}$. The size of a prediction set ranges from 0 (OOD rejection), to 1 (single-valued prediction), to somewhere in $\{2, \ldots, K-1\}$ (ambiguous observations), and ultimately to $K$ (ambiguity rejection). Unlike CRO, which only rejects normal observations, or SCOD, which mixes rejections of normal and OOD observations, our unified decision rule not only rejects hard normal observations but also effectively distinguishes them from OOD rejections.

**Objective Function.** In addition to the task of OOD detection, we aim to minimize the expected size of prediction set $\mathbb{E}_{\mathcal{Q}}[|\phi(\boldsymbol{X})|] = \sum_{k=1}^K \mathbb{E}_{\mathcal{Q}}[\mathbb{1}\{f_k(\boldsymbol{X}) \geq 0\}]$, subject to the class-specific error $\mathbb{E}_{\mathcal{Q}}[\mathbb{1}\{f_k(\boldsymbol{X}) < 0\} \mid Y = k] \leq \gamma, k \in [K]$, where $\gamma$ is prescribed by users due to the business needs. Denote $\mathcal{G}_k, k \in [K]$ as the index set of labeled data from class $k$ (with size $m_k$) and $\mathcal{G}_u$ as the index set of unlabeled data (with size $n$), we solve a data-driven optimization problem:

$$\min_{\boldsymbol{f} \in \mathcal{F}_L} \quad \frac{1}{nK} \sum_{i \in \mathcal{G}_u} \sum_{k=1}^K \ell_1(f_k(\boldsymbol{x}_i)) + C \sum_{k=1}^K \sum_{j \in \mathcal{G}_k} \frac{\omega_{k,j}}{m_k} \cdot \ell_{2,\gamma}(f_k(\boldsymbol{x}_j)) + J(\boldsymbol{f}), \tag{1}$$

where $\boldsymbol{f}(\boldsymbol{x}) := \mathbf{W}_L \widehat{\boldsymbol{\zeta}}(\boldsymbol{x}) - \boldsymbol{\rho}$ comes from the neural networks hypothesis class $\mathcal{F}_L$ with depth $L$. Here $\mathbf{W}_L$ is the weight matrix in the output layer of the network, $\widehat{\boldsymbol{\zeta}}(\boldsymbol{x})$ is the embedding learned from the penultimate layer, and $\boldsymbol{\rho} \in \mathbb{R}^K$ is the offset term. The regularization $J(\boldsymbol{f}) := \sum_{l=1}^L \frac{\lambda_l}{2} \|\mathbf{W}_l\|_F^2 + \sum_{k=1}^K \lambda_k'(-\rho_k)$ is used to confine the hypothesis class with parameters $\lambda_l, l \in [L]$ and $\lambda_k', k \in [K]$.

Instead of the 0-1 loss $\mathbb{1}\{u \geq 0\}$ in $\sum_{k=1}^K \mathbb{E}_{\mathcal{Q}}[\mathbb{1}\{f_k(\boldsymbol{X}) \geq 0\}]$, the first term in (1) measures the empirical prediction set size under a surrogate hinge loss $\ell_1(u) = [1 + u]_+ = \max\{0, 1 + u\}$. The second term in (1) aims to provide a non-negative upper bound of $\mathbb{E}_{\mathcal{Q}}[\mathbb{1}\{f_k(\boldsymbol{X}) < 0\} \mid Y = k] - \gamma$. To this end, notice that $\mathbb{1}\{u < 0\} - \gamma \leq [1 - \gamma - u]_+$ (see Fig. 2). By choosing $\ell_{2,\gamma}(u) = [1 - \gamma - u]_+$, minimizing the second term amounts to minimizing the excess empirical class-specific error beyond $\gamma$. When the loss $\ell_{2,\gamma}(\cdot)$ in the second term goes to 0, the empirical error tends to be less than $\gamma$. The tuning parameter $C$ balances the risk between the prediction set size and the misclassification rate. Due to the Assumption 1, we use the labeled data from $\mathcal{P}$ to quantify the empirical misclassification rate in the second term initially measured under distribution $\mathcal{Q}$. Lastly, similar to OCSVM (Schölkopf et al., 2000), together with the Gaussian kernel, $-\rho_k$ in the third term $J(\boldsymbol{f})$ in (1) penalizes the offset and often allows to exclude most atypical observations from acceptance regions (see the intuition and discussion in Section 3). To avoid negative values in the optimization stage, one may use $e^{-\rho_k}$ instead of $-\rho_k$ in the third term.

**Gaussian Kernel and Its Approximation.** OCSVM achieves anomaly detection by using the Gaussian kernel with offset penalization (like the third term in (1)). Nonetheless, it is difficult to recover the exact features after the kernel mapping in the context of neural networks as the resultant feature would be infinite-dimensional. Moreover, the Representer theorem (Kimeldorf & Wahba, 1971) suggests that the decision function is a linear combination of the kernel function evaluated at all the training data points. This can also be challenging in real business because each time we update the model, we only use a mini-batch (subset) of training data to avoid computation and memory burden.

To overcome the above difficulties, in the penultimate layer of the network (see Fig. 3), we use finite Random Fourier Features (Rahimi & Recht, 2007; Lu et al., 2016; Nguyen & Vien, 2018) to approximate the infinite-dimensional Gaussian kernel features. More concretely, we sample $D = m_{L-1}/2$ many independent frequencies $\boldsymbol{\xi}_j$ $(j = 1, \ldots, D)$ from the Gaussian distribution $\mathcal{N}(\boldsymbol{0}, \mathbf{I}_{m_{L-2}})$, where $m_{L-2}$ and $m_{L-1}$ are the widths of the $(L-2)$- and $(L-1)$-th layers, respectively. Then we let the mapped feature fed to the output layer be

$$\widehat{\boldsymbol{\zeta}}(\boldsymbol{x}) = \widehat{\boldsymbol{\zeta}}(\boldsymbol{z}_{\boldsymbol{x}}; \sigma) := D^{-1/2} \left( \sin(\sigma \boldsymbol{\xi}_1^\top \boldsymbol{z}_{\boldsymbol{x}}), \cos(\sigma \boldsymbol{\xi}_1^\top \boldsymbol{z}_{\boldsymbol{x}}), \ldots, \sin(\sigma \boldsymbol{\xi}_D^\top \boldsymbol{z}_{\boldsymbol{x}}), \cos(\sigma \boldsymbol{\xi}_D^\top \boldsymbol{z}_{\boldsymbol{x}}) \right)^\top, \tag{2}$$

where $\boldsymbol{z}_{\boldsymbol{x}}$ is the embedding of $\boldsymbol{x}$ output from the previous layer and the learnable parameter $\sigma$ relates the kernel's flexibility. Bochner's theorem (Rudin, 2017) shows that, for any $\boldsymbol{z}_{\boldsymbol{x}_i}, \boldsymbol{z}_{\boldsymbol{x}_j}$, the inner product

$\widehat{\boldsymbol{\zeta}}(\boldsymbol{z}_{\boldsymbol{x}_i};\sigma)^\top \widehat{\boldsymbol{\zeta}}(\boldsymbol{z}_{\boldsymbol{x}_j};\sigma)$ is an unbiased estimator to the true Gaussian kernel similarity $\exp\left(-\sigma^2\|\boldsymbol{z}_{\boldsymbol{x}_i} - \boldsymbol{z}_{\boldsymbol{x}_j}\|^2/2\right)$, where $\sigma^2$ plays a similar role as the bandwidth of Gaussian kernel in OCSVM. With the above kernel approximation, one can incorporate it into the neural network and use SGD optimization to efficiently update the model and hence scale the problem.

**Insight behind the Gaussian Kernel (Approximation) with Offset Penalization.** The Gaussian kernel and the formulation of the decision function $\boldsymbol{f}(\boldsymbol{x})$ with the penalization for $-\rho_k, k \in [K]$ in (1) are commonly used in (shallow) SVM-based anomaly detection (Schölkopf et al., 2000; Jumutc & Suykens, 2014; Shilton et al., 2020).

The term $-\rho_k$ in $f_k(\boldsymbol{x}) = \mathbf{W}_{L,k}\widehat{\boldsymbol{\zeta}}(\boldsymbol{x}) - \rho_k$ in $\boldsymbol{f}(\boldsymbol{x})$ typically comes with the term $-\rho_k$ in $J(\boldsymbol{f})$, where $\mathbf{W}_{L,k}$ denotes the $k$-th row of the matrix $\mathbf{W}_L$. Note that the feature mapping/embedding $\widehat{\boldsymbol{\zeta}}(\cdot)$ involved with Gaussian kernel approximation maps each input $\boldsymbol{x}$ to the sphere of a ball because of $\widehat{\boldsymbol{\zeta}}(\boldsymbol{x})^\top\widehat{\boldsymbol{\zeta}}(\boldsymbol{x}) = 1$. Under this context, as shown in Fig. 4, penalizing $-\rho_k$ during the optimization contributes to an increased distance (Schölkopf et al., 2000), $\frac{\rho_k}{\|\mathbf{W}_{L,k}\|_2}$, between the origin and the hyperplane $\mathbf{W}_{L,k}\widehat{\boldsymbol{\zeta}}(\boldsymbol{x}) = \rho_k$ in the feature space. This spatial shift effectively pushes the hyperplane outwards, resulting in a narrower acceptance region on

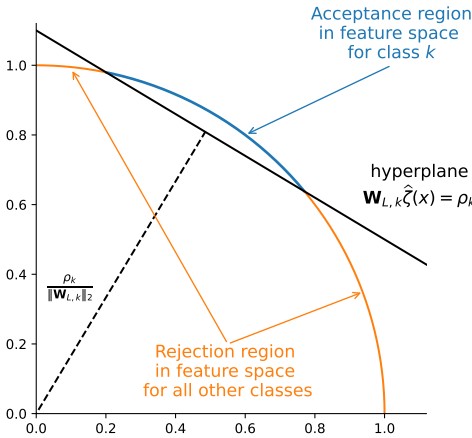

Figure 4: Illustration of Gaussian kernel (approximation) with offset penalization

the feature space sphere dedicated to the normal class $k$. It consequently creates more room for other classes (including the potential OOD class), possibly leading to an informative prediction and an improved OOD detection performance. To see the effectiveness of the approximated Gaussian Kernel with Offset Penalization, please see its ablation study in Fig. 7.

**Adaptive Weighted Loss.** The aforementioned loss function $\ell_{2,\gamma}(u)$ in (1) aims to provide a convex relaxation to $\mathbb{1}\{u < 0\} - \gamma$, whose expectation measures the class-specific error rate less a constant $\gamma$. However, there is a gap between $\mathbb{1}\{u < 0\} - \gamma$ and $\ell_{2,\gamma}(u)$: an observation $\boldsymbol{x}$ from the class $k$ outside of and far away from the center of the acceptance region $\mathcal{C}_k$ may incur a large loss $\ell_{2,\gamma}(\cdot)$, far larger than $1 - \gamma$. Additionally, the loss $\ell_{2,\gamma}(\cdot)$ may be non-zero even if a point falls in the correct acceptance region. In these cases, the loss $\ell_{2,\gamma}(\cdot)$ overestimates the true misclassification rate; as a consequence, we tend to get large acceptance regions $\mathcal{C}_k$, which inflates the prediction set size and degrades the OOD detection performance. To alleviate this potential issue, we allocate a weight to each observation to correct the overestimation. Specifically, we use a small weight for the observations with a large loss $\ell_{2,\gamma}(\cdot)$, by defining the weight as

$$\omega_{k,j} := \frac{1-\gamma}{1-\gamma-\check{f}_k(\boldsymbol{x}_j)} \cdot \mathbb{1}\{\check{f}_k(\boldsymbol{x}_j) < 0\} + \left(1 - \check{f}_k(\boldsymbol{x}_j)\right)^q \cdot \mathbb{1}\{0 \le \check{f}_k(\boldsymbol{x}_j) < 1-\gamma\}, \tag{3}$$

where $q > 0$. If $\check{f}_k(\boldsymbol{x}_j)$ in (3) is equal to $f_k(\boldsymbol{x}_j)$, then the weighted loss in (1) becomes

$$\omega_{k,j} \cdot \ell_{2,\gamma}(f_k(\boldsymbol{x}_j)) = \begin{cases} (1 - f_k(\boldsymbol{x}_j))^q \cdot \ell_{2,\gamma}(f_k(\boldsymbol{x}_j)), & 0 \le f_k(\boldsymbol{x}_j) < 1-\gamma \\ \left[\mathbb{1}\{f_k(\boldsymbol{x}_j) < 0\} - \gamma\right]_+, & \text{otherwise} \end{cases} . \tag{4}$$

This is illustrated in Fig. 2. Except for the middle interval, the weighted loss approximates $[\mathbb{1}\{u < 0\} - \gamma]_+$. In the middle interval, a large value of the parameter $q$ (we set $q = 5$) makes the weighted loss closer to 0. Here we use the classifier $\check{f}_k$ from the last iteration to approximate $f_k$.

## 4 Learning Theory

In this section, we show the convergence rates of the kernel approximation, the boundness of the true class-specific error rate, and the excess risk of the classification. We assume that $\|\boldsymbol{x}\|_2 \le c_0$.

Let $\mathcal{H}_{l,\kappa_l} := \{\boldsymbol{h}_l : \boldsymbol{h}_l(\boldsymbol{z}) = \mathcal{S}_l(\mathbf{W}_l\boldsymbol{z}), \mathbf{W}_l \in \mathbb{R}^{m_l \times p_l}, \|\mathbf{W}_l\|_F \le \kappa_l\}$ be the hypothesis class with bounded Frobenius norm in the $l$-th layer, where $\mathcal{S}_l$ is an (activation) function that is 1-Lipschitz continuous, e.g.,

ReLU$(\cdot), \cos(\cdot), \sin(\cdot)$, or the identity function. The input $\boldsymbol{z}$ is returned from the previous layer if $l > 1$ or equals $\boldsymbol{x}$ if $l = 1$. Let $\mathcal{F}_{L,\boldsymbol{\kappa}} := \{\boldsymbol{f} : \boldsymbol{h}_L \circ \cdots \circ \boldsymbol{h}_1, \boldsymbol{\rho} \geq \boldsymbol{0}, \boldsymbol{h}_l \in \mathcal{H}_{l,\kappa_l}, l \in [L]\}$ be a hypothesis class of deep dense neural networks (fused with a Gaussian kernel approximation) with depth $L$ (see Fig. 3) and $\boldsymbol{\kappa} = (\kappa_1, \ldots, \kappa_L)^\top$.

When the hypothesis class $\mathcal{F}_{L,\boldsymbol{\kappa}}$ is composited with certain loss functions, we introduce $\mathcal{F}_{L,\boldsymbol{\kappa}}^{sum,\ell_1} := \{\boldsymbol{x} \mapsto \sum_{k=1}^K \ell_1(f_k(\boldsymbol{x})) : \boldsymbol{f} = (f_1, \cdots, f_K) \in \mathcal{F}_{L,\boldsymbol{\kappa}}\}$ to denote a class, which involves applying the loss function $\ell_1(\cdot)$ element-wisely to $\boldsymbol{f}(\boldsymbol{x})$ and then summing the results. Furthermore, we define $\pi_k \circ \mathcal{F}_{L,\boldsymbol{\kappa}} := \{\boldsymbol{x} \mapsto f_k(\boldsymbol{x}) : \boldsymbol{f} = (f_1, \cdots, f_K) \in \mathcal{F}_{L,\boldsymbol{\kappa}}\}$ as another class that extracts the $k$-th component from $\boldsymbol{f}(\boldsymbol{x})$.

**Theorem 1.** *For each pair of observations $\boldsymbol{x}_i, \boldsymbol{x}_j$, let $\boldsymbol{z}_{\boldsymbol{x}_i}$ and $\boldsymbol{z}_{\boldsymbol{x}_j}$ be their embeddings learned at $(L-2)$-th layer, respectively. Denote $k_\sigma(\boldsymbol{z}_{\boldsymbol{x}_i}, \boldsymbol{z}_{\boldsymbol{x}_j}) = \exp\left(-\sigma^2\|\boldsymbol{z}_{\boldsymbol{x}_i} - \boldsymbol{z}_{\boldsymbol{x}_j}\|_2^2/2\right)$. For $\widehat{\boldsymbol{\zeta}}(\cdot; \sigma)$ defined in Eq. (2), with probability at least $1 - \delta$ over Gaussian frequency sampling, we have*

$$\left|\widehat{\boldsymbol{\zeta}}(\boldsymbol{z}_{\boldsymbol{x}_i}; \sigma)^\top \widehat{\boldsymbol{\zeta}}(\boldsymbol{z}_{\boldsymbol{x}_j}; \sigma) - k_\sigma(\boldsymbol{z}_{\boldsymbol{x}_i}, \boldsymbol{z}_{\boldsymbol{x}_j})\right| \leq \sqrt{\frac{4}{m_{L-1}} \log \frac{2}{\delta}} \text{ for fixed } \boldsymbol{z}_{\boldsymbol{x}_i} \text{ and } \boldsymbol{z}_{\boldsymbol{x}_j}, \tag{5}$$

$$\text{and} \quad \sup_{\boldsymbol{x}_i, \boldsymbol{x}_j} \left|\widehat{\boldsymbol{\zeta}}(\boldsymbol{z}_{\boldsymbol{x}_i}; \sigma)^\top \widehat{\boldsymbol{\zeta}}(\boldsymbol{z}_{\boldsymbol{x}_j}; \sigma) - k_\sigma(\boldsymbol{z}_{\boldsymbol{x}_i}, \boldsymbol{z}_{\boldsymbol{x}_j})\right| \leq \frac{4\sqrt{2}\sqrt{m_{L-2}}|\sigma|c_0}{\sqrt{m_{L-1}}} \prod_{l=1}^{L-2} \kappa_l + \sqrt{\frac{4}{m_{L-1}} \log \frac{2}{\delta}}. \tag{6}$$

For fixed embeddings $\boldsymbol{z}_{\boldsymbol{x}_i}$ and $\boldsymbol{z}_{\boldsymbol{x}_j}$ output from $(L-2)$-th layer, (5) implies that the cosine similarity between their mapped features in $(L-1)$-th layer converges to the true kernel similarity $k_\sigma(\boldsymbol{z}_{\boldsymbol{x}_i}, \boldsymbol{z}_{\boldsymbol{x}_j})$ at the rate of $\mathcal{O}(m_{L-1}^{-1/2})$ if $\sigma = \mathcal{O}(m_{L-2}^{-1/2})$. For any pair of inputs $\boldsymbol{x}_i, \boldsymbol{x}_j \in \mathcal{X}$, (6) shows that besides the error due to the finite sampling of frequencies, the dissimilarity of two points propagated throughout the course of the network also contributes to the kernel approximation error. If the input space $\mathcal{X}$ is compact, eventually, the error uniformly vanishes as the width of the penultimate layer goes to infinity.

Let $\mathcal{F}_{L,\boldsymbol{\kappa}}^+(\gamma, \nu) := \{\boldsymbol{f} \in \mathcal{F}_{L,\boldsymbol{\kappa}} : \mathbb{E}_{\mathcal{Q}}[\ell_{2,\gamma}(f_k(\boldsymbol{X})) \mid Y = k] \leq \nu, k \in [K]\}$ be a subspace where the population class-specific surrogate loss is bounded, and its empirical counterpart be $\widehat{\mathcal{F}}_{L,\boldsymbol{\kappa}}^+(\gamma, \nu) := \{\boldsymbol{f} \in \mathcal{F}_{L,\boldsymbol{\kappa}} : \frac{1}{n_k} \sum_{y_i=k} \ell_{2,\gamma}(f_k(\boldsymbol{x}_i)) \leq \nu, k \in [K]\}$. Without loss of generality, we set the adaptive weight in front of $\ell_{2,\gamma}$ to be 1. Theoretically, this simplification does not hurt our main theorems too much since it only affects the complexity of the loss function class by its Lipschitz constant. Then, by moving the second and third terms in Problem (1) to the constraint, we consider the below problem

$$\min_{\boldsymbol{f} \in \widehat{\mathcal{F}}_{L,\boldsymbol{\kappa}}^+(\gamma, \nu)} \frac{1}{nK} \sum_{i=1}^n \sum_{k=1}^K \ell_1(f_k(\boldsymbol{x}_i)). \tag{7}$$

The below theorem gives an upper bound to the true class-specific misclassification rate less the advertised value $\gamma$ measured under $\ell_{2,\gamma}$ loss.

**Theorem 2.** *Let $\hat{\boldsymbol{f}}$ be a solution to Problem (7) and $\vartheta_{n_k}(\delta) := 2\mathfrak{R}_{n_k}(\pi_k \circ \mathcal{F}_{L,\boldsymbol{\kappa}}) + r\sqrt{\frac{2}{n_k} \log \frac{2K}{\delta}}$, where the Rademacher complexity $\mathfrak{R}_{n_k}(\pi_k \circ \mathcal{F}_{L,\boldsymbol{\kappa}}) = \mathcal{O}(\frac{\log(\sqrt{n_k})}{\sqrt{n_k}})$ and $r = c_0 \prod_{l=1}^L \kappa_l$. With probability at least $1 - \delta$, simultaneously for all normal class $k \in [K]$, we have*

$$\mathbb{E}_{\mathcal{Q}}\left[\ell_{2,\gamma}(\hat{f}_k(\boldsymbol{X})) \mid Y = k\right] \leq \frac{1}{n_k} \sum_{y_i=k} \ell_{2,\gamma}(\hat{f}_k(\boldsymbol{x}_i)) + \vartheta_{n_k}(\delta).$$

Together with the fact $\mathbb{1}\{u < 0\} - \gamma \leq \ell_{2,\gamma}(u)$ and $\hat{\boldsymbol{f}} \in \widehat{\mathcal{F}}_{L,\boldsymbol{\kappa}}^+(\gamma, \nu)$, the above theorem indicates $\mathbb{P}_{\mathcal{Q}}[\hat{f}_k(\boldsymbol{X}) < 0 \mid Y = k] - \gamma \leq \nu + \vartheta_{n_k}(\delta)$, which suggests a way to control true misclassification rate. To bound the true misclassification rate by $\gamma$, one may use a more stringent tolerance, e.g., replace $\gamma$ in the loss function $\ell_{2,\gamma}$ by $\gamma - \theta$ where $\theta \geq \nu + \vartheta_{n_k}(\delta)$. This holds for a large enough dataset and with a large value of $C$ since a large value of $C$ corresponds to the small value of $\nu$, and $\vartheta_{n_k}(\delta) \to 0$ as $n_k \to \infty$.

Theorem 3 shows the classification risk, namely $\ell_1$-ambiguity, returned by an empirical minimizer converges to the least in the hypothesis class when the sample size of each normal class increases.

**Theorem 3.** *Let* $r, \vartheta_{n_k}(\delta)$ *take the forms as in Theorem 2, and* $\mathcal{R}_{\ell_1}(\hat{\boldsymbol{f}}) := \sum_{k=1}^{K} \mathbb{E}_{\mathcal{Q}}[\ell_1(\hat{f}_k(\boldsymbol{X}))]$ *be the* $\ell_1$*-ambiguity, where* $\hat{\boldsymbol{f}}$ *is an empirical minimizer of problem*

$$\min_{\boldsymbol{f} \in \widehat{\mathcal{F}}_{L,\boldsymbol{\kappa}}^+(\gamma - \nu - \vartheta^*, \nu)} \frac{1}{nK} \sum_{i=1}^{n} \sum_{k=1}^{K} \ell_1(f_k(\boldsymbol{x}_i)) \tag{8}$$

*and* $\vartheta^* = \max_k \vartheta_{n_k}(\delta)$. *With probability at least* $1 - 3\delta$, *we have*

*(1)* $\mathbb{P}_{\mathcal{Q}}[\hat{f}_k(\boldsymbol{X}) < 0 \mid Y = k] \leq \gamma$ *for* $k \in [K]$; *and*

*(2)* $\mathcal{R}_{\ell_1}(\hat{\boldsymbol{f}}) - \min_{\tilde{\nu} \in [0, \gamma - 2\vartheta^*]} \min_{\boldsymbol{f} \in \mathcal{F}_{L,\boldsymbol{\kappa}}^+(\gamma - \tilde{\nu}, \tilde{\nu})} \mathcal{R}_{\ell_1}(\boldsymbol{f}) \leq 12\Re_n(\mathcal{F}_{L,\boldsymbol{\kappa}}^{sum,\ell_1}) + 12\sqrt{K}r\sqrt{\frac{1}{2n}\log\frac{2}{\delta}}$.

By using a more stringent error tolerance and fine-tuning the parameter $C$, statement (1) shows the true misclassification rate of each normal class is below the advertised tolerance $\gamma$. Note that the second term in the L.H.S. of the statement (2) denotes the best performance over all those classifiers with the true misclassification rate bounded by $\gamma$ (here the interval for $\tilde{\nu}$ holds when $\vartheta^* \to 0$ as $n_k \to \infty$). Together with the fact that the Rademacher complexity $\Re_n(\mathcal{F}_{L,\boldsymbol{\kappa}}^{sum,\ell_1}) = \mathcal{O}(n^{-1/2}\log(n))$ vanishes as $n \to \infty$, Statement (2) implies the prediction $\hat{\phi}(\boldsymbol{x})$ returned by the DeepGPS approaches to the least $\ell_1$-ambiguity within the hypothesis class as $n_k \to \infty, k \in [K]$. For the explicit expressions of $\Re_{n_k}(\pi_k \circ \mathcal{F}_{L,\boldsymbol{\kappa}})$ and $\Re_n(\mathcal{F}_{L,\boldsymbol{\kappa}}^{sum,\ell_1})$, please direct to (16) and (21) in Appendix D.

## 5  Experiments

**Baselines.** DeepGPS is compared with baselines (GPS, BCOPS, and CDL) tailored to the task of set-valued classification with OOD detection. Since these baselines are shallow methods, to conduct a fair comparison, we also use the learned features from DeepGPS as the inputs for the baselines and refer to them as DeepGPS-based baselines, following the same regime in Ruff et al. (2020). We also compare with two SCOD methods, namely SIRC (Xia & Bouganis, 2022) and OpenMix (Zhu et al., 2023), which demonstrates the limitation of single-valued predictions in the current task.

**Datasets.** We deploy all methods on CIFAR-10, MNIST, and Fashion-MNIST datasets. In CIFAR-10, the normal classes are {*Bird*, *Cat*, *Deer*, *Dog*, *Frog*, *Horse*} and the OOD class comes from {*Airplane*, *Car*, *Ship*, *Truck*}. In MNIST, the normal classes are digits {*1*, *4*, *6*, *7*, *9*} and the OOD class consists of digits {*0*, *2*, *3*, *5*, *8*}. In Fashion-MNIST, the normal classes are {*Pullover*, *Dress*, *Coat*, *Sandal*, *Ankle boot*} and the OOD class is {*T-shirt*, *Trouser*, *Shirt*, *Sneaker*, *Bag*} . The values of $\gamma$ are prescribed as 0.05, 0.01, and 0.01 for CIFAR-10, MNIST, and Fashion-MNIST, respectively (the latter two tasks are relatively easy, hence a smaller $\gamma$). We split each original dataset into three sets: labeled set containing only normal classes to mimic distribution $\mathcal{P}$, unlabeled set mixing normal and OOD classes to mimic distribution $\mathcal{Q}$, and the test set. The first two are used to train the model, and the test set is to evaluate the performance.

**Network Architectures.** For backbone architectures, we use ResNet18 (He et al., 2016) on CIFAR-10, and LeNet-type CNNs (Ruff et al., 2020) on grayscale images MNIST and Fashion-MNIST. On top of the backbones, we add the head network (see Fig. 3) composed of 2 hidden layers with 500 units before the kernel approximation layer. Other hyper-parameters are discussed in Appendix A.

**Metrics.** We present the sample class-specific accuracy, the aligned OOD recall, and the aligned efficiency (see Appendix B for the details of alignment). Additionally, we report the AURec (area under the curve between the OOD recall and the accuracy) and AUEff (area under the curve between the efficiency and accuracy) to see the overall performance of a set-valued classifier around the neighborhood of $1 - \gamma$ (see the definitions in Appendix B). Intuitively, the higher the OOD recall and the higher the efficiency, the better the classifier. The bold numbers in Table 1 denote the best performances among all set-valued classifiers.

**Results.** The results of two SCOD methods are reported when the rate of incorrectly rejecting normal observations is around $\gamma$ by thresholding their score functions. As shown in Table 1, even though they maintain the highest (100%) efficiency, the overly-confident single-valued classification rule fails to guarantee the class-specific accuracy for normal classes. Contrastively, by providing plausible labels for certain

Table 1: Average performance metrics on CIFAR-10, MNIST, and Fashion-MNIST

| | | | DeepGPS | DeepGPS-based | | | GPS | BCOPS | CDL | SCOD | |
| | | | | GPS | BCOPS | CDL | | | | SIRC | OpenMix |
|---|---|---|---|---|---|---|---|---|---|---|---|
| **CIFAR-10** | Acc. | Bird | 96.7$_{\pm0.33}$ | 96.2$_{\pm0.4}$ | 94.3$_{\pm0.39}$ | 95.2$_{\pm0.79}$ | 96.5$_{\pm0.5}$ | 95.3$_{\pm0.3}$ | 94$_{\pm0.43}$ | 76.8$_{\pm0.7}$ | 75$_{\pm1.48}$ |
| | | Cat | 94.6$_{\pm0.46}$ | 93.3$_{\pm1.27}$ | 94.4$_{\pm0.41}$ | 94.7$_{\pm0.51}$ | 94.5$_{\pm0.68}$ | 94.8$_{\pm0.59}$ | 95.7$_{\pm0.03}$ | 62.6$_{\pm1.35}$ | 65.1$_{\pm1.07}$ |
| | | Deer | 97.1$_{\pm0.43}$ | 95.1$_{\pm0.56}$ | 95.5$_{\pm0.6}$ | 95.8$_{\pm0.56}$ | 96.4$_{\pm0.72}$ | 94.5$_{\pm0.28}$ | 96.2$_{\pm0.3}$ | 80.5$_{\pm1.07}$ | 75.9$_{\pm1.8}$ |
| | | Dog | 96.1$_{\pm0.5}$ | 94.8$_{\pm0.63}$ | 95.2$_{\pm0.48}$ | 94.8$_{\pm0.21}$ | 95.2$_{\pm0.77}$ | 95.9$_{\pm0.22}$ | 94.6$_{\pm0.51}$ | 69.2$_{\pm0.71}$ | 67.6$_{\pm0.99}$ |
| | | Frog | 95.8$_{\pm0.11}$ | 96.3$_{\pm0.45}$ | 95.6$_{\pm0.4}$ | 95.4$_{\pm0.23}$ | 94.7$_{\pm0.32}$ | 96.9$_{\pm0.26}$ | 95.6$_{\pm0.53}$ | 86.2$_{\pm0.54}$ | 84$_{\pm1.55}$ |
| | | Horse | 95.1$_{\pm0.61}$ | 94.2$_{\pm0.83}$ | 94.8$_{\pm0.44}$ | 95.4$_{\pm0.47}$ | 95.9$_{\pm0.25}$ | 95.4$_{\pm0.36}$ | 95.9$_{\pm0.26}$ | 84.4$_{\pm0.83}$ | 84.8$_{\pm0.94}$ |
| | Aligned OOD Recall | | **66.9**$_{\pm1.83}$ | 66.8$_{\pm1.33}$ | 63.7$_{\pm1.87}$ | 20.3$_{\pm4.55}$ | 19.8$_{\pm0.44}$ | 16.5$_{\pm0.24}$ | 0.2$_{\pm0.03}$ | 12.9$_{\pm0.49}$ | 14.4$_{\pm0.91}$ |
| | Aligned Efficiency | | **72.6**$_{\pm0.9}$ | 60$_{\pm1.85}$ | 67$_{\pm0.36}$ | 20.5$_{\pm2.55}$ | 16.8$_{\pm0.24}$ | 25.3$_{\pm0.09}$ | 10.3$_{\pm0.09}$ | 100$_{\pm0}$ | 100$_{\pm0}$ |
| | AURec | | 56.7$_{\pm3.1}$ | 57.8$_{\pm2.26}$ | **59.8**$_{\pm2.21}$ | 19.4$_{\pm4.38}$ | 18.4$_{\pm0.31}$ | 15.6$_{\pm0.33}$ | 0.2$_{\pm0.02}$ | / | / |
| | AUEff | | **65.1**$_{\pm0.9}$ | 55$_{\pm1.02}$ | 61.1$_{\pm0.29}$ | 19.8$_{\pm2.4}$ | 16.3$_{\pm0.18}$ | 23.9$_{\pm0.18}$ | 9.9$_{\pm0.1}$ | / | / |
| **MNIST** | Acc. | Digit 1 | 99.7$_{\pm0.1}$ | 99.1$_{\pm0.33}$ | 99.5$_{\pm0.04}$ | 99.2$_{\pm0.07}$ | 99.6$_{\pm0.02}$ | 99.4$_{\pm0.07}$ | 99.1$_{\pm0.12}$ | 99.3$_{\pm0.23}$ | 99.2$_{\pm0.16}$ |
| | | Digit 4 | 99.2$_{\pm0.18}$ | 99.3$_{\pm0.28}$ | 99.2$_{\pm0.17}$ | 99$_{\pm0.22}$ | 99.3$_{\pm0.26}$ | 99.5$_{\pm0.07}$ | 99.1$_{\pm0.12}$ | 98.5$_{\pm0.14}$ | 98.6$_{\pm0.13}$ |
| | | Digit 6 | 99.1$_{\pm0.04}$ | 98.6$_{\pm0.05}$ | 98.7$_{\pm0.15}$ | 98.4$_{\pm0.13}$ | 99.2$_{\pm0.23}$ | 98.6$_{\pm0.06}$ | 98.7$_{\pm0.17}$ | 99.1$_{\pm0.07}$ | 99.2$_{\pm0.1}$ |
| | | Digit 7 | 99.1$_{\pm0.18}$ | 99.4$_{\pm0.11}$ | 98.8$_{\pm0.21}$ | 98.8$_{\pm0.2}$ | 98.6$_{\pm0.14}$ | 98.3$_{\pm0.11}$ | 98.7$_{\pm0.16}$ | 98.6$_{\pm0.32}$ | 98$_{\pm0.11}$ |
| | | Digit 9 | 98.6$_{\pm0.14}$ | 98.9$_{\pm0.27}$ | 98.9$_{\pm0.1}$ | 99.1$_{\pm0.17}$ | 98.7$_{\pm0.16}$ | 98.5$_{\pm0.11}$ | 98.8$_{\pm0.18}$ | 97.4$_{\pm0.52}$ | 98.1$_{\pm0.13}$ |
| | Aligned OOD Recall | | **90.4**$_{\pm1.27}$ | 75.8$_{\pm1.9}$ | 79$_{\pm1.59}$ | 59.2$_{\pm4.34}$ | 74.1$_{\pm3.46}$ | 73.3$_{\pm0.88}$ | 8.8$_{\pm0.55}$ | 23.8$_{\pm3.05}$ | 39.1$_{\pm2.31}$ |
| | Aligned Efficiency | | **99.7**$_{\pm0.06}$ | 96.2$_{\pm0.47}$ | 96.8$_{\pm0.56}$ | 85.7$_{\pm1.45}$ | 88$_{\pm2.19}$ | 90.4$_{\pm0.4}$ | 37.4$_{\pm1.2}$ | 100$_{\pm0}$ | 100$_{\pm0}$ |
| | AURec | | **83.1**$_{\pm2.47}$ | 67.2$_{\pm2.84}$ | 72$_{\pm1.67}$ | 54.1$_{\pm4.09}$ | 66.5$_{\pm2.28}$ | 67.1$_{\pm0.89}$ | 8.1$_{\pm0.45}$ | / | / |
| | AUEff | | **97.5**$_{\pm0.43}$ | 92.3$_{\pm1.52}$ | 94$_{\pm0.66}$ | 82.8$_{\pm1.55}$ | 84$_{\pm1.68}$ | 87.8$_{\pm0.47}$ | 35.3$_{\pm0.95}$ | / | / |
| **Fashion-MNIST** | Acc. | Pullover | 98.9$_{\pm0.05}$ | 99$_{\pm0.15}$ | 98.8$_{\pm0.14}$ | 99.2$_{\pm0.19}$ | 99.1$_{\pm0.12}$ | 98.9$_{\pm0.07}$ | 99.1$_{\pm0.1}$ | 93.6$_{\pm0.24}$ | 92.2$_{\pm0.42}$ |
| | | Dress | 98.6$_{\pm0.16}$ | 99.1$_{\pm0.11}$ | 98.6$_{\pm0.31}$ | 98.8$_{\pm0.2}$ | 98.1$_{\pm0.5}$ | 98.3$_{\pm0.08}$ | 99.2$_{\pm0.11}$ | 94$_{\pm1.1}$ | 95.3$_{\pm0.33}$ |
| | | Coat | 99.1$_{\pm0.15}$ | 99.1$_{\pm0.38}$ | 99.4$_{\pm0.18}$ | 99.4$_{\pm0.06}$ | 98.5$_{\pm0.55}$ | 99.3$_{\pm0.07}$ | 98.9$_{\pm0.13}$ | 88.5$_{\pm0.35}$ | 90.5$_{\pm0.58}$ |
| | | Sandal | 99.7$_{\pm0.08}$ | 99.3$_{\pm0.29}$ | 99.2$_{\pm0.12}$ | 99.5$_{\pm0.06}$ | 98.7$_{\pm0.44}$ | 98.7$_{\pm0.11}$ | 98.8$_{\pm0.1}$ | 99.2$_{\pm0.09}$ | 99.4$_{\pm0.1}$ |
| | | Ankle boot | 99.3$_{\pm0.12}$ | 99.2$_{\pm0.24}$ | 98.9$_{\pm0.12}$ | 98.9$_{\pm0.2}$ | 98.9$_{\pm0.18}$ | 99$_{\pm0.1}$ | 99.2$_{\pm0.16}$ | 99.4$_{\pm0.1}$ | 99.3$_{\pm0.07}$ |
| | Aligned OOD Recall | | **59.2**$_{\pm1.52}$ | 48.4$_{\pm1.96}$ | 56.1$_{\pm1.29}$ | 32.7$_{\pm1.97}$ | 40.5$_{\pm3.38}$ | 54$_{\pm0.72}$ | 4.9$_{\pm0.25}$ | 4.4$_{\pm0.52}$ | 9.3$_{\pm0.8}$ |
| | Aligned Efficiency | | **91.1**$_{\pm0.11}$ | 88.9$_{\pm0.51}$ | 90$_{\pm0.28}$ | 81.3$_{\pm1.18}$ | 84.5$_{\pm0.61}$ | 88$_{\pm0.22}$ | 43$_{\pm0.58}$ | 100$_{\pm0}$ | 100$_{\pm0}$ |
| | AURec | | 49.7$_{\pm1.43}$ | 40.3$_{\pm0.58}$ | **50.5**$_{\pm1}$ | 30.6$_{\pm1.84}$ | 35.6$_{\pm3.19}$ | 49.3$_{\pm0.4}$ | 4.4$_{\pm0.22}$ | / | / |
| | AUEff | | **86.8**$_{\pm0.32}$ | 83.9$_{\pm1.81}$ | **86.8**$_{\pm0.39}$ | 78.2$_{\pm1.16}$ | 80.8$_{\pm0.7}$ | 85.6$_{\pm0.09}$ | 41$_{\pm0.41}$ | / | / |

observations, set-valued classification controls the accuracy and returns cautious decisions for those classes of interest. Without representation learning, three shallow set-valued baselines exhibit inferior results on both OOD recall and efficiency (or their AUCs). With the learned features from DeepGPS, the OOD recall and efficiency of the DeepGPS-based baselines are significantly improved, but they are still not as good as DeepGPS overall. In contrast, besides controlling the class-specific accuracy, the proposed end-to-end DeepGPS also exhibits high OOD recall and the highest efficiency under the prescribed accuracy.

Notice that the performance of set-valued classification is driven by the prescribed accuracy level. Intuitively, aiming for a higher prescribed accuracy tends to decrease the prediction efficiency (i.e., larger prediction sets) and lower the OOD recall. This trade-off is shown through the curves in Fig. 5, where the top row illustrates the change in OOD recall across different accuracy thresholds, and the bottom row demonstrates the corresponding changes in prediction efficiency. Both the values of AURec and AUEff in Table 1 are computed based on these curves. Overall, our proposed method DeepGPS returns efficient prediction sets while maintaining competing OOD recall.

Fig. 6 illustrates some examples of OOD-rejected images ($|\hat{\phi}(\boldsymbol{x})| = 0$) in the first row and ambiguous images ($1 < |\hat{\phi}(\boldsymbol{x})| < K$) in the second row with the predicted labels shown in the bracket. For instance, the ambiguous cat (row 2, col. 3 in Fig. 6a) is classified with the set {*Dog, Deer, Cat, Horse*} (possibly due to its upright legs and hair color), while the cat (row 1, col. 3) is rejected as an OOD point because of its rare posture. A plane from the OOD class is classified as {*Bird, Frog*} since the shape and the green color confuse the classifier. A car in the first row is rightfully rejected as an OOD point since its red profile is unlike any other normal class. These set-valued decisions allow for further inspection in the presence of ambiguity to reduce the risk of misclassification.

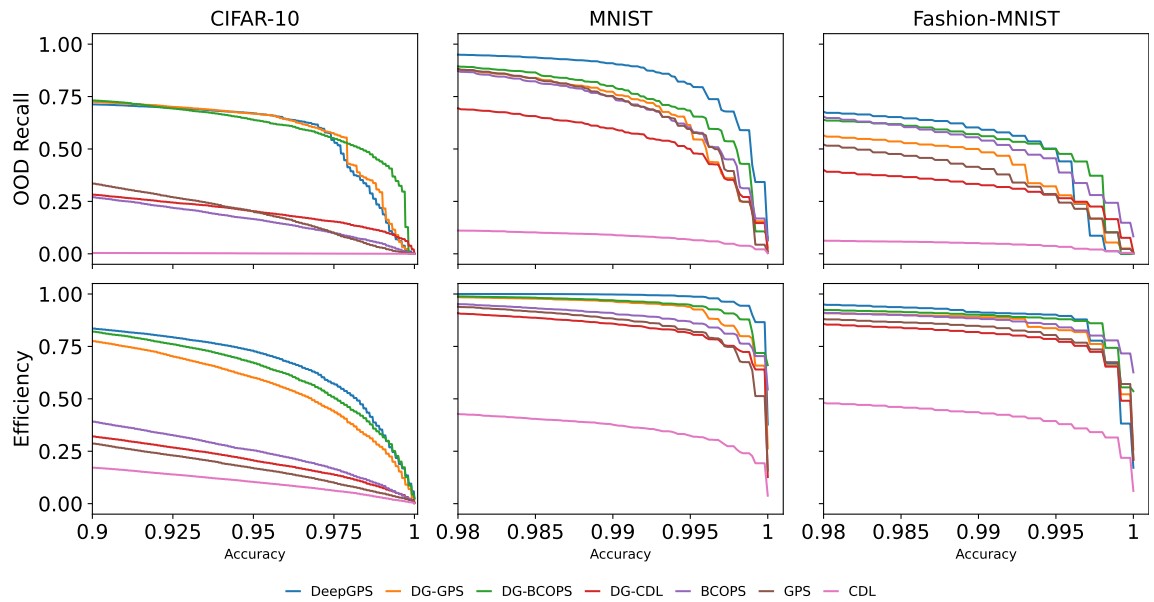

Figure 5: Trade-off between OOD recall (and efficiency) with accuracy. Methods associated with "DG-" denote the baselines trained on the features obtained from DeepGPS.

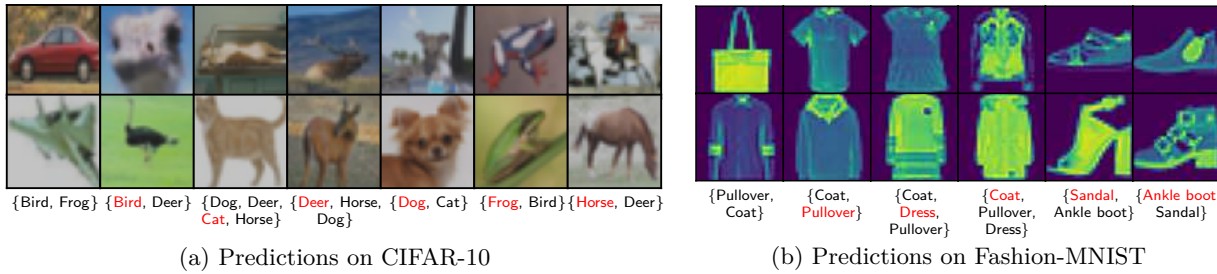

(a) Predictions on CIFAR-10        (b) Predictions on Fashion-MNIST

Figure 6: For each panel, images in the same column come from the same class (OOD class or a certain normal class). The first row in each panel refers to OOD-rejected images; the second row refers to ambiguous images with the predicted labels shown in the bracket. Note that the ground truth class is shown in red. Images in the first column in each panel are from the OOD class.

**Ablation for Kernel Approximation with Offset Penalization.** As was discussed in Section 3, the Gaussian Kernel Approximation in the penultimate layer allows us to scale the computation. In addition, we now conduct an ablation study for the Kernel Approximation with Offset Penalization (KAOP) technique to show its impact on the acceptance regions. By setting the number of output neurons in the backbone network to two, we can visualize the MNIST and Fashion-MNIST datasets and the closure and tightness of the acceptance regions. As shown in Fig. 7, DeepGPS with the KAOP (right panel) outputs closed and tighter acceptance regions than the one without KAOP (left panel). Some acceptance regions in the left panel are fairly large, possibly not even closed. This means that potentially more OOD points are wrongly accepted and decisions can be more ambiguous.

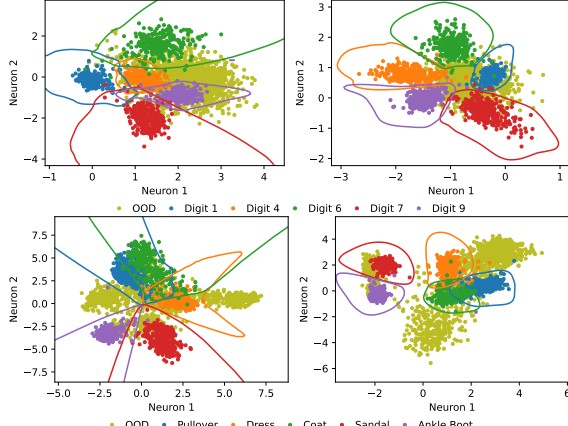

Figure 7: Acceptances regions for MNIST (top) and Fashion-MNIST (bottom), without (left) and with (right) the KAOP technique.

Fig. 8 exhibits the performance of DeepGPS with and without using the KAOP technique and using the weighted loss. When solely comparing Bar1 with Bar2 (or Bar3 with Bar4), we can see that the method without KAOP either fails to balance the trade-off between OOD recalls and efficiencies, or returns the lowest OOD recalls and efficiencies. In CIFAR-10, for example, DeepGPS without KAOP yields an extremely low efficiency. For this dataset, the KAOP technique significantly improves the efficiency at the cost of a slightly smaller OOD recall. For both MNIST and Fashion-MNIST, KAOP improves both the efficiency and the OOD recall in general.

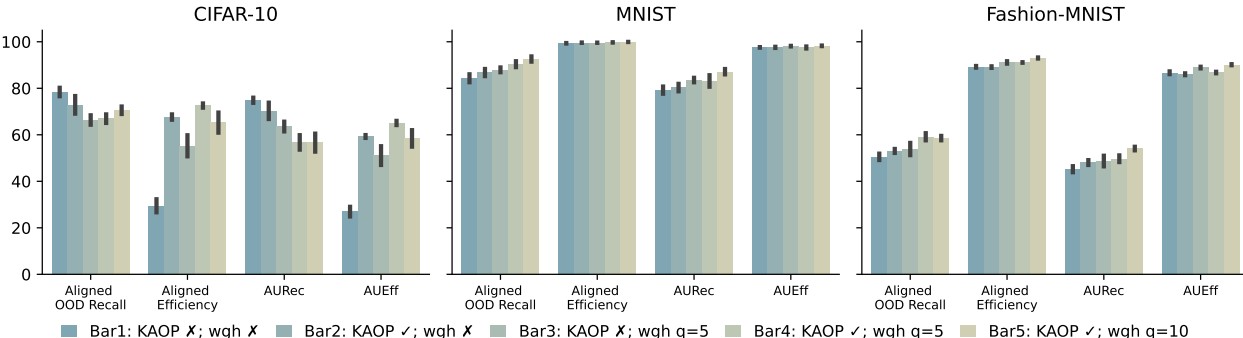

Figure 8: Ablation studies for KAOP and weighted loss. ✓ denotes a technique is deployed, while ✗ denotes not. $q = 5$ or $10$ denotes the parameter value in the weighted loss.

**Ablation for Weighted Loss.** Similar to the impact of KAOP, the weighted loss provides a trade-off between prediction efficiency and OOD recall for the CIFAR-10; see the comparison between Bar1 and Bar3 (or Bar2 and Bar4). For the MNIST and Fashion-MNIST data, both efficiency and OOD recall are improved due to the use of the weighted loss. Different values of the parameter $q$ in the weighted loss lead to a similar effect on the efficiency and OOD recall, when compared to the method without using the weighted loss (see Bar4 and Bar5 when compared to Bar2).

Overall, Fig. 8 shows that with both KAOP and weighted loss, we tend to obtain compact acceptance regions that have high prediction efficiency and OOD recall.

**Sensitivity for OOD Proportion.** In addition, we include the sensitivity study (in Appendix C.3) to explore DeepGPS with varying proportions of OOD data in the target distribution $\mathcal{Q}$. The efficiency remains relatively stable across different OOD proportions in all three datasets.

# 6 Conclusion

Conventional single-valued predictions make decisions without guaranteed confidence for interested classes. Moreover, current set-valued classification methods have sub-optimal performances on large and complex datasets. To address these limitations, we propose an end-to-end DeepGPS method. Empirically, besides detecting OOD points, DeepGPS provides reliable control over class-specific accuracy for normal classes, offering cautious yet informative decisions to mitigate risks. Theoretically, we show that DeepGPS minimizes the prediction set size under the prescribed accuracy. These support the effectiveness of DeepGPS in scenarios where misclassification may have severe consequences, and hence highly accurate predictions are desired.

The DeepGPS network provides scalable set-valued classification with OOD detection. The kernel approximation allows Deep Neural Networks to tap into the good learning property of the kernels. This, along with the offset penalization, facilitates the construction of closed acceptance regions. The usage of weighted loss further renders more compact acceptance regions. As per the resulting predictions, DeepGPS differentiates between OOD-rejected observations (i.e., $|\hat{\phi}(\boldsymbol{x})| = 0$), which were not considered in the CRO method, from difficult observations (i.e., $2 \leq |\hat{\phi}(\boldsymbol{x})| \leq K$). This distinction has not been well-explicitly explored in the OOD detection and SCOD literature.

While the current implementation learns a shared $\sigma$ value in Random Fourier Features, it is important to note that this approach may overlook the heterogeneity among different classes, potentially resulting in larger

acceptance regions for certain classes and hence degrading the performance. Future work could involve the design of a new architecture that enables the learning of class-specific $\sigma$ values. Additionally, our proposed method leverages unlabeled datasets. Expanding the framework to operate in an online learning mode, allowing for incremental updates with limited data availability, presents an intriguing avenue for future exploration. Lastly, proposing a method that even controls the OOD recall with a theoretical guarantee would be another intriguing topic. However, to our best knowledge, this further relies on the assumption of the OOD data (Liu et al., 2018; Fang et al., 2021), which might be challenging in practice.

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

## Appendix

## A    Details of Experiments

All deep methods consistently use the same backbone and head network with the same dimension of weight matrices in each layer. Experiments run over 150 epochs with batch size 512. The ResNet18 (He et al., 2016) architecture is the same as the one in PyTorch and LeNet-type CNNs are identical to Ruff et al. (2020). The training set (labeled set and unlabeled set as mentioned in Section 5) is further split with a ratio 9:1 into training data for learning $\boldsymbol{f}$ and validation data for tuning parameters. The average and standard errors of performances are reported after 5 runs.

**DeepGPS.** We use the Adam optimizer (Kingma & Ba, 2014) with learning rate $lr = 10^{-4}$ and $(\beta_1, \beta_2) = (0.999, 0.999)$. Additionally, we set the weight decay in Adam as $10^{-4}$ and set $\lambda_l = 0, l \in [L]$ in the objective function Eq. (1) to implicitly regularize weight matrices instead of explicitly imposing a penalty on the first term in $J(\boldsymbol{f})$. In contrast, the parameter $\lambda'_k = 1, k \in [K]$, same as in OCSVM (Schölkopf et al., 2000), is to explicitly penalize the offset term in $J(\boldsymbol{f})$. The tuning parameter $C$ is determined such that the prediction set is smallest on the unlabeled part in the validation data when the misclassification rate is close to $\gamma$ on the labeled part in the validation data.

For the below set-valued baselines (or DeepGPS-based variants), we conduct the split-conformal (Vovk et al., 2005; Lei et al., 2015) and search for the optimal tuning parameters on the validation data (Lei, 2014). In particular, let $\hat{f}_k$ be the score function returned by a baseline and $\hat{\tau}_k$ be a $\gamma \times 100\%$ quantile of $\{f_k(\boldsymbol{x}_j)\}_{j \in \mathcal{G}_k^{val}}$, where $\mathcal{G}_k^{val}$ denotes the index set of the observations from class $k$ in the validation data. Then the optimal parameter is chosen when the prediction set size $\sum_{j \in \mathcal{G}_u^{val}} \sum_{k=1}^{K} \mathbb{1}\{\hat{f}_k(\boldsymbol{x}_j) \geq \hat{\tau}_k\}$ is minimized on the unlabeled points in the validation data, where $\mathcal{G}_u^{val}$ denotes the index set of unlabeled observations in the validation data.

**GPS.** We search the value of penalty in the GPS method from the grid $\{0.1, 1, 10\}$, and use the median of pairwise Euclidean distance among observations as the bandwidth in its Gaussian kernel.

**BCOPS.** The maximum depth of the tree is searched from $\{10, 30, 50\}$. Minimum samples to split an internal node, minimum samples at a leaf node, and the number of trees are searched from $\{5, 10\}, \{4, 6\}$, and $\{50, 100\}$, respectively.

**CDL.** We search the value of bandwidth in an even grid with length 5, starting from $\hat{\sigma}_{(1)} \times (\frac{4}{(p+2)n})^{1/(p+4)}$ to $\hat{\sigma}_{(p)} \times (\frac{4}{(p+2)n})^{1/(p+4)}$, based on Silverman's rule-of-thumb bandwidth estimator (Silverman, 2018), where $n$ is the sample size, $p$ is the dimension of the feature and $\hat{\sigma}_{(1)}$ and $\hat{\sigma}_{(p)}$ are the minimum and maximum standard deviations across all dimension of features.

**SIRC.** By following Xia & Bouganis (2022), we use the score function consisting of maximum softmax probability and the $l_1$ norm of the embedding in the penultimate layer, where the former is to separate easy normal observations from both difficult normal observations and OOD observations while the latter is to distinguish normal observations from OOD observations.

**OpenMix.** This method requires OOD exposure data, where we use the Gaussian noise as mentioned in Zhu et al. (2023). In this method, the score function is the max logits returned from the output layer.

## B    Evaluation Metrics for Set-valued Classification

Note that the prediction performance is driven by the user-defined class-specific accuracy $1 - \gamma$. There exists a trade-off among accuracy, OOD detection performance, and efficiency. Intuitively, a higher prescribed accuracy leads to lower OOD detection performance and efficiency.

In our evaluation, we report several key metrics for assessing the performance from different perspectives. Let $\mathcal{G}_{te}$ be the index set of the test set. We report the sample class-specific accuracy

$$\frac{\sum_{j \in \mathcal{G}_{te}} \mathbb{1}\{Y_j = k \text{ and } Y_j \in \hat{\phi}(\boldsymbol{X}_j)\}}{\sum_{j \in \mathcal{G}_{te}} \mathbb{1}\{Y_j = k\}} \times 100\%, \ k \in [K]$$

to measure the accuracy of class predictions for normal classes; the aligned OOD recall

$$\text{Rec}(1 - \gamma) := \frac{\sum_{j \in \mathcal{G}_{te}} \mathbb{1}\{Y_j = \text{OOD and } |\hat{\phi}(\boldsymbol{X}_j)| = 0\}}{\sum_{j \in \mathcal{G}_{te}} \mathbb{1}\{Y_j = \text{OOD}\}} \times 100\%$$

to evaluate the ability to correctly identify OOD samples; and the aligned efficiency

$$\text{Eff}(1 - \gamma) := 1 - \frac{1}{K - 1} \cdot \left[ \frac{\sum_{j \in \mathcal{G}_{te}} \mathbb{1}\{Y_j \neq \text{OOD}\} \cdot |\hat{\phi}(\boldsymbol{X}_j)|}{\sum_{j \in \mathcal{G}_{te}} \mathbb{1}\{Y_j \neq \text{OOD}\}} - 1 \right]_+ \times 100\%$$

to show how the classifier distinguishes between normal observations. The first term in $[\cdot]_+$ in $\text{Eff}(1 - \gamma)$ denotes the average prediction set size on the normal observations, ranging from 0 to $K$.

The above two aligned metrics are obtained by adjusting the thresholds in such a way that a set-valued classifier exactly achieves the sample accuracy to be the prescribed value $1 - \gamma$ for each class, regardless of the goodness of the original sample class-specific accuracy. This strategy helps us to conduct a relatively fair comparison for set-valued classifiers since sample OOD recall and sample efficiency are affected by the sample class-specific accuracy.

In SIRC and OpenMix methods, to obtain the aligned OOD recall, we set the threshold for the score functions such that the error rate of incorrectly rejecting normal observations is $\gamma$. This strategy results in both single-valued and set-valued rules exhibiting a similar overall error rate of rejecting normal observations, mitigating potential disparities in our comparison. Notably, the single-valued prediction has 100% efficiency.

To see the overall performances, one may consider two new metrics, namely, AURec (area under the curve between OOD recall and accuracy) and AUEff (area under the curve between efficiency and accuracy). These two metrics are defined near the neighborhood of the prescribed accuracy $1 - \gamma$, i.e., from $1 - 2\gamma$ to 1:

$$\text{AURec} := \frac{1}{2\gamma} \int_{1-2\gamma}^{1} \text{Rec}(t) dt, \ \text{and} \ \text{AUEff} := \frac{1}{2\gamma} \int_{1-2\gamma}^{1} \text{Eff}(t) dt.$$

It is important to note that these overall performance metrics (even though defined near the neighborhood of $1 - \gamma$) have limitations as they overlook the specific accuracy value the user prescribed. Additionally, both AURec and AUEff are calculated when all normal classes attain every accuracy value within the integral region, which alludes that these two metrics are not applicable for single-valued classification, leaving them blank for both SIRC and OpenMix methods in our reports.

## C  Extra Experiments

### C.1  Set-valued Classification on CIFAR-100

We train the set-valued classification with ResNet50 backbone on the CIFAR-100 dataset with 20 coarser class labels. Here 10 normal classes consist of {*aquatic mammals*, *flowers*, *fruit and vegetables*, *natural outdoor scenes*, *omnivores and herbivores*, *medium-sized mammals*, *invertebrates*, *reptiles*, *small mammals*, *trees*}, while the remaining 10 classes form as the OOD class. Table 2 shows that almost all the methods can control the prescribed accuracy on all normal classes. With the representation learning, DeepGPS, DeepGPS-based GPS, BCOPS, and CDL outperform the vanilla GPS, BCOPS, and GPS. Additionally, the proposed DeepGPS has a better trade-off between OOD recall and efficiency.

Table 2: Average performance metrics for CIFAR-100 ($\gamma = 0.05$)

| CIFAR-100 | | DeepGPS | DeepGPS-based | | | GPS | BCOPS | CDL |
|---|---|---|---|---|---|---|---|---|
| | | | GPS | BCOPS | CDL | | | |
| | aquatic mammals | $98_{\pm 0.51}$ | $96.6_{\pm 0.3}$ | $94.4_{\pm 1.14}$ | $95.2_{\pm 0.8}$ | $96.9_{\pm 0.42}$ | $94.3_{\pm 0.61}$ | $95.6_{\pm 0.75}$ |
| | flowers | $97_{\pm 0.73}$ | $95.3_{\pm 1.09}$ | $94.7_{\pm 0.64}$ | $94.5_{\pm 0.85}$ | $95.8_{\pm 0.65}$ | $96.4_{\pm 0.83}$ | $93.1_{\pm 0.53}$ |
| | fruit and vegetables | $97_{\pm 0.54}$ | $91.8_{\pm 1.06}$ | $95.4_{\pm 0.54}$ | $95.1_{\pm 0.61}$ | $95.3_{\pm 0.54}$ | $95.4_{\pm 0.3}$ | $94.6_{\pm 0.94}$ |
| | natural outdoor scenes | $96.9_{\pm 1.64}$ | $94.2_{\pm 0.73}$ | $94.4_{\pm 1.04}$ | $94.6_{\pm 0.66}$ | $94.1_{\pm 0.54}$ | $95.8_{\pm 0.56}$ | $95.8_{\pm 0.22}$ |
| Acc. | omnivores and herbivores | $96.7_{\pm 0.46}$ | $96_{\pm 0.39}$ | $94.8_{\pm 0.8}$ | $96_{\pm 0.69}$ | $94.8_{\pm 1.54}$ | $95.8_{\pm 1.02}$ | $95_{\pm 0.32}$ |
| | medium-sized mammals | $96.6_{\pm 0.59}$ | $96_{\pm 0.22}$ | $95.6_{\pm 0.26}$ | $96.5_{\pm 0.54}$ | $94.9_{\pm 1.14}$ | $95.5_{\pm 0.68}$ | $96.1_{\pm 0.59}$ |
| | invertebrates | $94.5_{\pm 1}$ | $95.8_{\pm 1.75}$ | $95.1_{\pm 0.57}$ | $95.8_{\pm 1.12}$ | $96.5_{\pm 0.67}$ | $95.4_{\pm 0.28}$ | $94.6_{\pm 0.6}$ |
| | reptiles | $95.2_{\pm 1.11}$ | $96.1_{\pm 0.67}$ | $95.1_{\pm 0.93}$ | $94.6_{\pm 0.59}$ | $95.3_{\pm 0.53}$ | $94.3_{\pm 0.48}$ | $93_{\pm 0.69}$ |
| | small mammals | $95_{\pm 0.66}$ | $94.7_{\pm 0.87}$ | $95.7_{\pm 0.54}$ | $95.9_{\pm 0.52}$ | $91.8_{\pm 1.62}$ | $94_{\pm 0.64}$ | $94_{\pm 0.55}$ |
| | trees | $96.6_{\pm 1.11}$ | $93_{\pm 1.59}$ | $94.2_{\pm 1.12}$ | $95.1_{\pm 0.66}$ | $95.6_{\pm 0.82}$ | $95.4_{\pm 0.18}$ | $94.4_{\pm 0.67}$ |
| Aligned OOD Recall | | $\mathbf{43.9}_{\pm 1.49}$ | $1.8_{\pm 0.95}$ | $8.8_{\pm 1.35}$ | $6.3_{\pm 0.98}$ | $0_{\pm 0.03}$ | $0.1_{\pm 0.01}$ | $0_{\pm 0.01}$ |
| Aligned Efficiency | | $47.4_{\pm 3.64}$ | $53.9_{\pm 1.49}$ | $\mathbf{62.6}_{\pm 0.78}$ | $29.7_{\pm 1.96}$ | $33.7_{\pm 0.33}$ | $35.1_{\pm 0.31}$ | $11.5_{\pm 0.26}$ |
| AUODR | | $\mathbf{38.4}_{\pm 0.96}$ | $3.4_{\pm 0.99}$ | $8.5_{\pm 1.05}$ | $6.6_{\pm 0.9}$ | $0.1_{\pm 0.06}$ | $0.1_{\pm 0.01}$ | $0_{\pm 0.01}$ |
| AUEff | | $41_{\pm 2.44}$ | $48.8_{\pm 1.14}$ | $\mathbf{56.9}_{\pm 0.85}$ | $29_{\pm 1.32}$ | $32_{\pm 0.16}$ | $33.1_{\pm 0.25}$ | $11.1_{\pm 0.19}$ |

## C.2 Class-specific Accuracy Control in Ablation Studies

Fig. 9 exhibits the accuracy control in the ablation studies on three datasets, where the height of each bar represents the average sample accuracy and the black vertical segment on the top of each bar denotes the standard error. Additionally, red dashed horizontal lines denote the prescribed accuracy level of $1 - \gamma$. In this figure, we can see that the DeepGPS method effectively controls the class-specific accuracy across various techniques employed in the ablation studies. The metrics of prediction efficiency and OOD detection performances are displayed in Fig. 8.

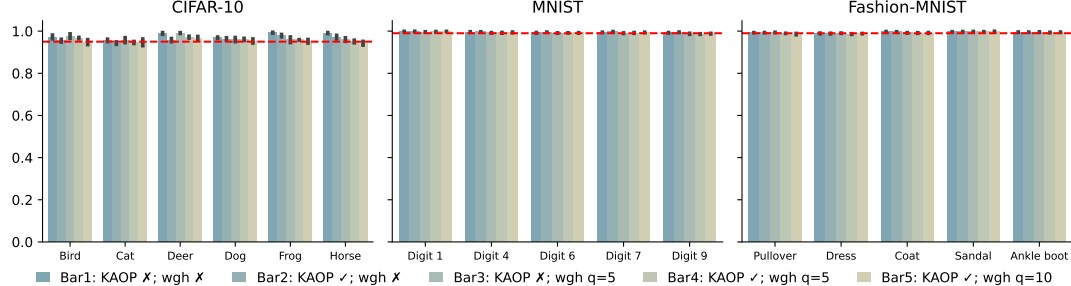

Figure 9: Accuracy control in ablation studies for KAOP and weighted loss. ✓ denotes a technique is deployed, while ✗ denotes not. $q = 5$ or $10$ denotes the parameter value in the weighted loss.

## C.3 Sensitivity for OOD Proportion

The results presented in Table 1 are based on OOD proportions of 4/10, 5/10, and 5/10 for CIFAR-10, MNIST, and Fashion-MNIST, respectively (due to the number of sub-classes chosen for the OOD class). In

Table 3: Sensitivity study for varied OOD proportions on CIFAR-10 ($\gamma = 0.05$)

| | Acc. | | | | | | Aligned OOD Recall | Aligned Efficiency | AURec | AUEff |
|---|---|---|---|---|---|---|---|---|---|---|
| | Bird | Cat | Deer | Dog | Frog | Horse | | | | |
| 0.2 | $95.4_{\pm 0.59}$ | $95.5_{\pm 0.34}$ | $96.8_{\pm 0.6}$ | $96.4_{\pm 0.54}$ | $97_{\pm 0.38}$ | $95.3_{\pm 1.21}$ | $15.8_{\pm 2.15}$ | $71.9_{\pm 0.87}$ | $14.8_{\pm 2.01}$ | $66.7_{\pm 0.68}$ |
| 0.3 | $95_{\pm 1.24}$ | $94.4_{\pm 0.75}$ | $95.9_{\pm 1.01}$ | $96.1_{\pm 0.74}$ | $96_{\pm 0.66}$ | $94.7_{\pm 1.29}$ | $33.8_{\pm 11.71}$ | $72.4_{\pm 1.33}$ | $32.2_{\pm 10.16}$ | $64.6_{\pm 1.29}$ |
| 0.8 | $96_{\pm 0.78}$ | $95.3_{\pm 0.88}$ | $96.8_{\pm 0.88}$ | $96_{\pm 0.57}$ | $96.5_{\pm 0.83}$ | $95.7_{\pm 0.66}$ | $70.7_{\pm 5.08}$ | $70.3_{\pm 1.13}$ | $66_{\pm 4.7}$ | $62.8_{\pm 0.74}$ |
| 0.9 | $96.5_{\pm 0.69}$ | $94.3_{\pm 1.26}$ | $97_{\pm 0.73}$ | $95.7_{\pm 0.8}$ | $96.7_{\pm 0.69}$ | $96_{\pm 0.81}$ | $73.6_{\pm 1.56}$ | $69.3_{\pm 1.84}$ | $70.4_{\pm 2.16}$ | $62_{\pm 1.36}$ |

this section, we include the sensitivity study to explore the performance of DeepGPS under another four different values, i.e., 0.2, 0.3, 0.8, and 0.9, of OOD proportion in three different datasets. As shown in Tables 3 to 5, the class-specific accuracies are well-controlled. The metrics of the efficiency, remain relatively stable across different OOD proportions in all three datasets. When it comes to the OOD detection performance, in general, the higher OOD proportion helps to improve the OOD detection performance.

Table 4: Sensitivity study for varied OOD proportions on MNIST ($\gamma = 0.01$)

| | Acc. | | | | | Aligned OOD Recall | Aligned Efficiency | AURec | AUEff |
|---|---|---|---|---|---|---|---|---|---|
| | Digit 1 | Digit 4 | Digit 6 | Digit 7 | Digit 9 | | | | |
| 0.2 | 99.9±0.02 | 99.4±0.09 | 99.3±0.13 | 99.2±0.16 | 99.2±0.11 | 68.2±2.2 | 99.8±0.06 | 60.1±2.92 | 97.6±0.31 |
| 0.3 | 99.9±0.06 | 99.4±0.1 | 99.3±0.13 | 99.1±0.2 | 98.9±0.09 | 80.4±1.5 | 99.7±0.08 | 70.2±2.07 | 97.2±0.26 |
| 0.8 | 99.7±0.05 | 99.4±0.07 | 99.1±0.08 | 99.1±0.22 | 98.8±0.14 | 89.4±1.59 | 99.5±0.12 | 82.4±1.65 | 97.2±0.32 |
| 0.9 | 99.7±0.04 | 99.4±0.04 | 99±0.1 | 99.3±0.16 | 98.7±0.1 | 88.5±1.82 | 99.5±0.12 | 81.1±2.57 | 97±0.46 |

Table 5: Sensitivity study for varied OOD proportions on Fashion-MNIST ($\gamma = 0.01$)

| | Acc. | | | | | Aligned OOD Recall | Aligned Efficiency | AURec | AUEff |
|---|---|---|---|---|---|---|---|---|---|
| | Pullover | Dress | Coat | Sandal | Ankle boot | | | | |
| 0.2 | 99.3±0.09 | 98.7±0.15 | 99.3±0.08 | 99.7±0.07 | 99.4±0.12 | 19.3±1.59 | 92.4±0.27 | 17.7±1.2 | 87.5±0.27 |
| 0.3 | 99±0.15 | 98.8±0.17 | 99±0.2 | 99.4±0.05 | 99.4±0.11 | 33.1±4.52 | 91.4±0.1 | 30±3.5 | 86.6±0.63 |
| 0.8 | 99.1±0.04 | 98.6±0.19 | 99.5±0.11 | 99.6±0.09 | 99.4±0.09 | 59.5±1.86 | 91.3±0.16 | 46.6±1 | 85.3±0.77 |
| 0.9 | 99±0.04 | 98.7±0.19 | 99.6±0.11 | 99.6±0.1 | 99.5±0.05 | 57.1±2.26 | 90.9±0.2 | 47.3±1.27 | 86.5±0.44 |

## C.4  Alternative Loss Functions

In this section, we test the performance of DeepGPS on the CIFAR-10 with different surrogate losses, i.e., exponential and logistic loss functions. Table 6 shows that the alternative two loss functions significantly sacrifice the prediction efficiency. This is mainly due to both logistic $\log(1 + \exp(-u))$ and exponential loss $\exp(-u)$ having non-zero loss value when $u > 0$, which forces the prediction sets to include more labels such that the empirical loss values are below than the error bound $\gamma$ on average. In contrast, the hinge loss used in Table 1 returns smaller prediction sets without sacrificing too much OOD recall.

Furthermore, we once considered the sigmoid function $\frac{1}{1+\exp(u)}$ as a candidate loss function which is closer to 0-1 loss than the hinge loss; however, the gradient vanishing with $u$ away from 0 hinders the training progress.

Table 6: Average performance on CIFAR-10 ($\gamma = 0.05$) under alternative loss functions

| Loss Functions | Acc. | | | | | | Aligned OOD Recall | Aligned Efficiency | AUODR | AUEff |
|---|---|---|---|---|---|---|---|---|---|---|
| | Bird | Cat | Deer | Dog | Frog | Horse | | | | |
| logistic | 96.7±0.68 | 97.7±0.58 | 98.5±0.53 | 98.3±0.73 | 98.6±0.51 | 98.2±0.53 | 83.1±0.28 | 34.1±6.37 | 71.7±3.12 | 33.7±5.11 |
| exponential | 97.3±0.68 | 97.9±0.54 | 99±0.45 | 99.1±0.21 | 99.1±0.22 | 98.4±0.41 | 84.8±1.51 | 36.2±5.3 | 70.7±2.23 | 35.9±4.2 |
| hinge (in Table 1) | 96.7±0.33 | 94.6±0.46 | 97.1±0.43 | 96.1±0.5 | 95.8±0.11 | 95.1±0.61 | 66.9±1.83 | 72.6±0.9 | 56.7±3.1 | 65.1±0.9 |

# D    Proof of Theorems

For a matrix $\mathbf{W}_l \in \mathbb{R}^{m_l \times p_l}$ in $l$-th layer, $l \in [L]$, if $\|\mathbf{W}_\ell\|_F \leq \kappa_l$, thus we have $\|\mathbf{W}_l\| \leq \kappa_l$ and $\|\mathbf{W}_l^\top\|_{2,1} \leq \sqrt{m_l}\kappa_l := b_l$, where $\|\cdot\|$ is the operator norm and $\|\cdot\|_{2,1}$ is the sum of Euclidean norms of the matrix columns.

To prove Theorem 1, we need the notion of Rademacher complexity to measure the richness of a hypothesis class. Following this, we show the Rademacher complexity of a particular hypothesis class with Lemma 1, which will be used in the subsequent proofs.

**Definition 1.** *Let $\mathcal{G}$ be a class of real-valued functions $g : \mathcal{X} \mapsto \mathbb{R}$. The Rademacher complexity $\mathfrak{R}_n(\mathcal{G})$ and the empirical Rademacher complexity $\widehat{\mathfrak{R}}_n(\mathcal{G})$ of the hypothesis class $\mathcal{G}$ are respectively defined as*

$$\mathfrak{R}_n(\mathcal{G}) := \mathbb{E}\left[\sup_{g\in\mathcal{G}} \frac{1}{n}\sum_{j=1}^{n} \sigma_j g(\boldsymbol{x}_j)\right], \quad \widehat{\mathfrak{R}}_n(\mathcal{G}) := \mathbb{E}\left[\sup_{g\in\mathcal{G}} \frac{1}{n}\sum_{j=1}^{n} \sigma_j g(\boldsymbol{x}_j) \mid \{\boldsymbol{x}_j\}_{j=1}^{n}\right],$$

*where $\sigma_j, j = 1, \ldots, n$ are Rademacher random variables which independently and uniformly take values from $\{-1, +1\}$, and $\boldsymbol{x}_j, j = 1, \ldots, n$ are sample points.*

**Remark 2.** *In Definition 1, the first expectation is taken with respect to both Rademacher random variables and sample points, while the second expectation is only taken with respect to Rademacher random variables by conditioning on sample points. Given this, we have $\mathfrak{R}_n(\mathcal{G}) = \mathbb{E}[\widehat{\mathfrak{R}}_n(\mathcal{G})]$.*

**Lemma 1.** *(Liang, 2016) Let $\tilde{\boldsymbol{x}}_i \in \mathbb{R}^p$ and assume $\mathbb{E}[\|\tilde{\boldsymbol{x}}_i\|_2^2] \leq c_{\tilde{\boldsymbol{x}}}^2$ holds for all $i = 1, \ldots, n$. The Rademacher complexity of the inner product operation with constrained $\ell_2$ norm on a dataset with sample size $n$ is*

$$\mathfrak{R}_n\left(\tilde{\boldsymbol{x}} \mapsto \tilde{\boldsymbol{x}}^\top \boldsymbol{\beta} : \|\boldsymbol{\beta}\|_2 \leq b_{\boldsymbol{\beta}}, \boldsymbol{\beta} \in \mathbb{R}^p\right) \leq \frac{c_{\tilde{\boldsymbol{x}}} b_{\boldsymbol{\beta}}}{\sqrt{n}}.$$

*Proof.*

$$
\begin{aligned}
\mathfrak{R}_n\left(\tilde{\boldsymbol{x}} \mapsto \tilde{\boldsymbol{x}}^\top \boldsymbol{\beta} : \|\boldsymbol{\beta}\|_2 \leq b_{\boldsymbol{\beta}}, \boldsymbol{\beta} \in \mathbb{R}^p\right) &= \frac{1}{n}\mathbb{E}\left[\sup_{\boldsymbol{\beta}:\|\boldsymbol{\beta}\|_2\leq b_{\boldsymbol{\beta}}} \sum_{i=1}^{n} \sigma_i \tilde{\boldsymbol{x}}_i^\top \boldsymbol{\beta}\right] = \frac{1}{n}\mathbb{E}\left[\sup_{\boldsymbol{\beta}:\|\boldsymbol{\beta}\|_2\leq b_{\boldsymbol{\beta}}} \boldsymbol{\beta}^\top \sum_{i=1}^{n} \sigma_i \tilde{\boldsymbol{x}}_i\right] \\
&\leq \frac{b_{\boldsymbol{\beta}}}{n}\mathbb{E}\left[\|\sum_{i=1}^{n} \sigma_i \tilde{\boldsymbol{x}}_i\|_2\right] \qquad\qquad \text{Cauchy-Schwartz inequality} \\
&= \frac{b_{\boldsymbol{\beta}}}{n}\mathbb{E}\left[\sqrt{\|\sum_{i=1}^{n} \sigma_i \tilde{\boldsymbol{x}}_i\|_2^2}\right] \leq \frac{b_{\boldsymbol{\beta}}}{n}\sqrt{\mathbb{E}\left[\|\sum_{i=1}^{n} \sigma_i \tilde{\boldsymbol{x}}_i\|_2^2\right]} \quad \sqrt{\cdot} \text{ is concave} \\
&\leq \frac{b_{\boldsymbol{\beta}}}{n}\sqrt{\mathbb{E}\left[\sum_{i=1}^{n} \|\tilde{\boldsymbol{x}}_i\|_2^2\right]} \leq \frac{b_{\boldsymbol{\beta}} \cdot c_{\tilde{\boldsymbol{x}}}}{\sqrt{n}}.
\end{aligned}
$$

$\square$

The main idea in the proof of Theorem 1 is to treat sampled the Gaussian frequencies $\boldsymbol{\xi}_j \in \mathbb{R}^{m_{L-2}}, j = 1, \ldots, D$ as "data points", and find the Rademacher complexity of certain mapping applied on these $\boldsymbol{\xi}_j$.

**Proof of Theorem 1**: Inequality (5) can be obtained by using the Hoeffding's inequality. To show (6), let $\Delta_{\boldsymbol{z}} := \boldsymbol{z}_{\boldsymbol{x}_i} - \boldsymbol{z}_{\boldsymbol{x}_j}$ and

$$
\begin{aligned}
K_D &:= \sup_{\Delta_{\boldsymbol{z}}} \widehat{\boldsymbol{\zeta}}(\boldsymbol{z}_{\boldsymbol{x}_i}; \sigma)^\top \widehat{\boldsymbol{\zeta}}(\boldsymbol{z}_{\boldsymbol{x}_j}; \sigma) - \exp\left(-\frac{\sigma^2 \|\boldsymbol{z}_{\boldsymbol{x}_i} - \boldsymbol{z}_{\boldsymbol{x}_j}\|^2}{2}\right) \\
&= \sup_{\Delta_{\boldsymbol{z}}} \frac{1}{D}\sum_{j=1}^{D} \cos\left(\sigma \boldsymbol{\xi}_j^\top \Delta_{\boldsymbol{z}}\right) - \exp\left(-\frac{\sigma^2 \|\Delta_{\boldsymbol{z}}\|^2}{2}\right).
\end{aligned}
$$

By McDiarmid's inequality (the bounded difference is $\frac{2}{D}$), with the probability at least $1 - \delta$, we have

$$K_D \leq \mathbb{E}[K_D] + \sqrt{\frac{2}{D} \log \frac{2}{\delta}} \leq 2 \cdot \mathfrak{R}_D(\{\boldsymbol{\xi} \mapsto \cos(\sigma \boldsymbol{\xi}^\top \Delta_{\boldsymbol{z}})\}) + \sqrt{\frac{2}{D} \log \frac{2}{\delta}} \tag{9}$$

$$\leq 2 \cdot \mathfrak{R}_D(\{\boldsymbol{\xi} \mapsto \sigma \boldsymbol{\xi}^\top \Delta_{\boldsymbol{z}}\}) + \sqrt{\frac{2}{D} \log \frac{2}{\delta}} \tag{10}$$

$$\leq 2\sqrt{m_{L-2}} \frac{|\sigma| \cdot 2c_0 \prod_{l=1}^{L-2} \kappa_l}{\sqrt{D}} + \sqrt{\frac{2}{D} \log \frac{2}{\delta}}, \tag{11}$$

where $\mathbb{E}[K_D] \leq 2 \cdot \mathfrak{R}_D(\{\boldsymbol{\xi} \mapsto \cos(\sigma \boldsymbol{\xi}^\top \Delta_{\boldsymbol{z}})\})$ in (9) holds due to Ma (2022, Theorem 4.13); (10) holds by using Talagrand's lemma (Mohri et al., 2018) and the fact that $\cos(\cdot)$ is a 1-Lipschitz function. As for (11), it is true since

$$\mathfrak{R}_D(\{\boldsymbol{\xi} \mapsto \sigma \boldsymbol{\xi}^\top \Delta_{\boldsymbol{z}}\}) \leq \frac{\sqrt{m_{L-2}} \cdot |\sigma| \cdot \sup \|\Delta_{\boldsymbol{z}}\|_2}{\sqrt{D}} \qquad \text{because of Lemma 1 and } \boldsymbol{\xi}_j \sim \mathcal{N}(\mathbf{0}, \mathbf{I}_{m_{L-2}})$$

$$\leq \frac{\sqrt{m_{L-2}} \cdot |\sigma|}{\sqrt{D}} \cdot (\prod_{l=1}^{L-2} \kappa_l) \cdot \|\boldsymbol{x}_i - \boldsymbol{x}_j\|_2 \tag{12}$$

$$\leq \frac{\sqrt{m_{L-2}} \cdot |\sigma| \cdot 2c_0}{\sqrt{D}} \cdot \prod_{l=1}^{L-2} \kappa_l,$$

where (12) holds because $\boldsymbol{z}_i, \boldsymbol{z}_j$ are outputs from the neural network up to $(L - 2)$-th layer, which is also a Lipschitz mapping (with Lipschitz constant $\prod_{l=1}^{L-2} \kappa_l$) evaluated on $\boldsymbol{x}_i, \boldsymbol{x}_j$. The last inequality holds due to the assumption of $\|\boldsymbol{x}_i\|_2 \leq c_0$ mentioned in Section 4.

To derive Theorems 2 and 3, we need the notion of the covering number with metric entropy, which is another approach to show the capacity of a hypothesis class. The covering number and the (empirical) Rademacher complexity can be bridged by Dudley's Theorem. Additionally, we introduce Lemmas 2 and 3 that are critical to deriving our theorems.

**Definition 2.** *The covering number of a hypothesis class $\mathcal{F}$ with respect to a metric $\rho : \mathcal{F} \times \mathcal{F} \mapsto \mathbb{R}$ is defined as the smallest number of balls with radius $\varepsilon$ covering $\mathcal{F}$, i.e.,*

$$\mathcal{N}(\varepsilon, \mathcal{F}, \rho) := \min\{N : \exists \{f_1, f_2, \cdots, f_N\} \subset \mathcal{F} \subset \cup_{i=1}^N \mathcal{B}_\rho(f_i, \varepsilon)\},$$

*where the ball with center $f_i$ and radius $\varepsilon$ under the metric $\rho$ is defined as $\mathcal{B}_\rho(f_i, \varepsilon) := \{f : \rho(f_i, f) \leq \varepsilon\}$. Additionally, the metric entropy is defined as the logarithm of the covering number, i.e., $\log \mathcal{N}(\varepsilon, \mathcal{F}, \rho)$.*

**Remark 3.** *In particular, throughout this article, the distance between two functions $f_i$ and $f$ under the metric $\rho = L_2(\mathbb{P}_n)$ is defined as*

$$\rho(f_i, f) = L_2(\mathbb{P}_n)(f_i, f) = \int \|f_i - f\|_2^2 d\mathbb{P}_n := \sqrt{\frac{1}{n} \sum_{i=1}^n \|f_i(\boldsymbol{x}_j) - f(\boldsymbol{x}_j)\|_2^2},$$

*where $\boldsymbol{x}_j, j = 1, \ldots, n$ are sample points. Here $f_i(\cdot)$ and $f(\cdot)$ could be a vector output.*

**Lemma 2.** *(Bartlett et al., 2017; Ma, 2022) Let $\mathcal{H}'_{l,b_l} = \{\boldsymbol{z} \mapsto \mathbf{W}\boldsymbol{z} : \mathbf{W} \in \mathbb{R}^{m_l \times p_l}, \|\mathbf{W}^\top\|_{2,1} \leq b_l\}$ and $\|\boldsymbol{z}_i\|_2 \leq c_{l-1}, i = 1, \ldots, n$. The metric entropy*

$$\log \mathcal{N}(\varepsilon_l, \mathcal{H}'_{l,b_l}, L_2(\mathbb{P}_n)) \leq \frac{b_l^2 c_{l-1}^2}{\varepsilon_l^2} \ln(2m_l p_l).$$

**Remark 4.** *Slightly different from Bartlett et al. (2017); Ma (2022), for notation simplicity, we do not apply ceiling operation on $\frac{b_l^2 c_{l-1}^2}{\varepsilon_l^2}$ in Lemma 2. This is not a big deal because we can remove the ceiling operation by adjusting the value of $\varepsilon_l$ to obtain the ceiling integer.*

**Remark 5.** *Lemma 2 can immediately yield* $\log \mathcal{N}\left(\varepsilon_l, \mathcal{H}_{l,\kappa_l}, L_2(\mathbb{P}_n)\right) \leq \frac{b_l^2 c_{l-1}^2}{\varepsilon_l^2} \ln(2m_l p_l)$ *with* $b_l = \sqrt{m_l}\kappa_l$ *as defined at very beginning.*

**Lemma 3.** *Ma (2022) Assume the input* $\boldsymbol{z}_{l-1}$ *in l-th layer of the network architecture* $\mathcal{F}_{L,\boldsymbol{\kappa}}$ *satisfies* $\|\boldsymbol{z}_{l-1}\|_2 \leq c_{l-1}$. *If the metric entropy of the hypothesis class in l-th layer has an upper bound* $\log \mathcal{N}\left(\varepsilon_l, \mathcal{H}_{l,\kappa_l}, L_2(\mathbb{P}_n)\right) \leq g(\varepsilon_l, c_{l-1})$, *where* $\varepsilon_l$ *is the radius of covering balls for* $\mathcal{H}_{l,\kappa_l}$, *there exists an* $\varepsilon$-*covering of* $\mathcal{F}_{L,\boldsymbol{\kappa}}$ *such that the metric entropy*

$$\log \mathcal{N}\left(\varepsilon, \mathcal{F}_{L,\boldsymbol{\kappa}}, L_2(\mathbb{P}_n)\right) \leq \sum_{l=1}^{L} g(\varepsilon_l, c_{l-1}).$$

**Theorem 4.** *Let* $r := c_0 \cdot \prod_{l=1}^{L} \kappa_l$. *For any Lipschitz mapping* $\psi : \mathbb{R}^K \mapsto \mathbb{R}$ *with Lipschitz constant* $c_\psi$, *under the assumptions in Lemma 3, the Rademacher complexity of the hypothesis class* $\psi \circ \mathcal{F}_{L,\boldsymbol{\kappa}}$ *on the dataset with sample size n is*

$$\mathfrak{R}_n(\psi \circ \mathcal{F}_{L,\boldsymbol{\kappa}}) \leq \frac{4}{\sqrt{n}} + \frac{12 \cdot c_\psi \cdot r \log(r\sqrt{n})}{\sqrt{n}} \cdot \left[\sum_{l=1}^{L} \left(m_l \ln(2m_l p_l)\right)^{\frac{1}{3}}\right]^{\frac{3}{2}}.$$

*Proof.* By using Remark 5 and Lemma 3, we have

$$\log \mathcal{N}(\varepsilon, \mathcal{F}_{L,\boldsymbol{\kappa}}, L_2(\mathbb{P}_n)) \leq \sum_{l=1}^{L} \frac{b_l^2 c_{l-1}^2}{\varepsilon_l^2} \ln(2m_l p_l). \tag{13}$$

Our goal now is to bound the metric entropy $\log \mathcal{N}(\varepsilon, \mathcal{F}_{L,\boldsymbol{\kappa}}, L_2(\mathbb{P}_n))$ directly involved with $\varepsilon$ instead of $\varepsilon_l$ in (13). As shown in the proof of Lemma 3 in Ma (2022), there exists a covering for $\mathcal{F}_{L,\boldsymbol{\kappa}}$ with radius $\varepsilon = \sum_{l=1}^{L} \varepsilon_l \prod_{j=l+1}^{L} \kappa_j$. Then, through the Holder's inequality, we have

$$\sum_{l=1}^{L} \left(b_l c_{l-1} \sqrt{\ln(2m_l p_l)} \prod_{j=l+1}^{L} \kappa_j\right)^{\frac{2}{3}} = \sum_{l=1}^{L} \left(\frac{b_l c_{l-1}}{\varepsilon_l} \sqrt{\ln(2m_l p_l)}\right)^{\frac{2}{3}} \cdot \left(\varepsilon_l \prod_{j=l+1}^{L} \kappa_j\right)^{\frac{2}{3}}$$

$$\leq \left[\sum_{l=1}^{L} \frac{b_l^2 c_{l-1}^2}{\varepsilon_l^2} \ln(2m_l p_l)\right]^{\frac{1}{3}} \cdot \left[\underbrace{\sum_{l=1}^{L} \left(\varepsilon_l \prod_{j=l+1}^{L} \kappa_j\right)}_{\varepsilon}\right]^{\frac{2}{3}}$$

$$\Rightarrow \quad \left[\sum_{l=1}^{L} \left(b_l c_{l-1} \sqrt{\ln(2m_l p_l)} \prod_{j=l+1}^{L} \kappa_j\right)^{\frac{2}{3}}\right]^3 \frac{1}{\varepsilon^2} \leq \sum_{l=1}^{L} \frac{b_l^2 c_{l-1}^2}{\varepsilon_l^2} \ln(2m_l p_l)$$

and equality can be achieved when we choose

$$\varepsilon_l = \frac{\varepsilon (b_l/\kappa_l)^{1/2}(\ln(2m_l p_l))^{1/4}}{\prod_{j=l+1}^{L} \kappa_l} \cdot \left(\sum_{l=1}^{L} (\frac{b_l}{\kappa_l}\sqrt{\ln(2m_l p_l)})^{2/3}\right)^{-3/2} \cdot \left(\sum_{l=1}^{L} \frac{b_l}{\kappa_l}\sqrt{\ln(2m_l p_l)}\right)^{1/2}.$$

Hence, the upper bound in (13) can be re-expressed as

$$\sum_{l=1}^{L} \frac{b_l^2 c_{l-1}^2}{\varepsilon_l^2} \ln(2m_l p_l) = \left[\sum_{l=1}^{L} \left(b_l c_{l-1} \sqrt{\ln(2m_l p_l)} \prod_{j=l+1}^{L} \kappa_j\right)^{\frac{2}{3}}\right]^3 \frac{1}{\varepsilon^2}$$

$$= \prod_{l=1}^{L} \kappa_l^2 \left[\sum_{l=1}^{L} \left(b_l c_{l-1} \sqrt{\ln(2m_l p_l)} \prod_{j=1}^{l} \kappa_j^{-1}\right)^{\frac{2}{3}}\right]^3 \frac{1}{\varepsilon^2} = \underbrace{c_0^2 \prod_{l=1}^{L} \kappa_l^2}_{r^2} \left[\sum_{l=1}^{L} (\sqrt{m_l} \ln(2m_l p_l))^{\frac{2}{3}}\right]^3 \frac{1}{\varepsilon^2}$$

where the last equality holds due to $c_l = c_0 \cdot \prod_{j=1}^{l} \kappa_j$ (composition of Lipschitz continuous functions) and $b_l = \sqrt{m_l} \kappa_l$. Thereby Eq. (13) concludes

$$\log \mathcal{N}(\varepsilon, \mathcal{F}_{L,\kappa}, L_2(\mathbb{P}_n)) \leq \frac{r^2}{\varepsilon^2} \cdot \left[ \sum_{l=1}^{L} (m_l \ln(2m_l p_l))^{\frac{1}{3}} \right]^3.$$

Together with the Lipschitz transformation property in the covering number (Park & Muandet, 2023), i.e., $\mathcal{N}(\varepsilon, \psi \circ \mathcal{F}_{L,\kappa}, L_2(\mathbb{P}_n)) \leq \mathcal{N}(\varepsilon/c_\psi, \mathcal{F}_{L,\kappa}, L_2(\mathbb{P}_n))$, the above further indicates

$$\log \mathcal{N}(\varepsilon, \psi \circ \mathcal{F}_{L,\kappa}, L_2(\mathbb{P}_n)) \leq \frac{c_\psi^2 r^2}{\varepsilon^2} \cdot \left[ \sum_{l=1}^{L} (m_l \ln(2m_l p_l))^{\frac{1}{3}} \right]^3,$$

According to the Localized Dudley's Theorem (Ma, 2022), we have

$$\widehat{\mathfrak{R}}_n(\psi \circ \mathcal{F}_{L,\kappa}) \leq 4\alpha + 12 \int_\alpha^r \sqrt{\frac{\log(\varepsilon, \psi \circ \mathcal{F}_{L,\kappa}, L_2(\mathbb{P}_n))}{n}} d\varepsilon$$

$$\leq \frac{4}{\sqrt{n}} + \frac{12 c_\psi \cdot r \log(r\sqrt{n})}{\sqrt{n}} \cdot \left[ \sum_{l=1}^{L} (m_l \ln(2m_l p_l))^{\frac{1}{3}} \right]^{\frac{3}{2}}$$

by choosing $\alpha = \frac{1}{\sqrt{n}}$. Therefore,

$$\mathfrak{R}_n(\psi \circ \mathcal{F}_{L,\kappa}) = \mathbb{E}\left[ \widehat{\mathfrak{R}}_n(\psi \circ \mathcal{F}_{L,\kappa}) \right] \leq \frac{4}{\sqrt{n}} + \frac{12 c_\psi \cdot r \log(r\sqrt{n})}{\sqrt{n}} \cdot \left[ \sum_{l=1}^{L} (m_l \ln(2m_l p_l))^{\frac{1}{3}} \right]^{\frac{3}{2}}.$$

$\square$

**Proof of Theorem 2**: Define $G_{n_k} := \sup_{f \in \widehat{\mathcal{F}}_{L,\kappa}^+(\gamma,\nu)} \mathbb{E}[\ell_{2,\gamma}(f_k(\boldsymbol{X})) \mid Y = k] - \frac{1}{n_k} \sum_{y_i=k} \ell_{2,\gamma}(f_k(\boldsymbol{x}_i))$. By the McDiarmid's inequality (the bounded difference is $\frac{2r}{n_k}$), with probability at least $1 - \frac{\delta}{K}$, we have

$$G_{n_k} \leq \mathbb{E}[G_{n_k}] + 2r\sqrt{\frac{1}{2n_k} \log \frac{2K}{\delta}} \leq 2 \cdot \mathfrak{R}_{n_k}(\ell_{2,\gamma} \circ \pi_k \circ \widehat{\mathcal{F}}_{L,\kappa}^+(\gamma,\nu)) + r\sqrt{\frac{2}{n_k} \log \frac{2K}{\delta}}, \tag{14}$$

where $\pi_k$ denotes the $k$-th coordinate projection.

Note that both loss function $\ell_{2,\gamma}$ and projection operator $\pi_k$ are 1-Lipschitz mappings, and $\mathfrak{R}_{n_k}(\ell_{2,\gamma} \circ \pi_k \circ \widehat{\mathcal{F}}_{L,\kappa}^+(\gamma,\nu)) \subset \mathfrak{R}_{n_k}(\ell_{2,\gamma} \circ \pi_k \circ \mathcal{F}_{L,\kappa}^+(\gamma,\nu))$ because of $\widehat{\mathcal{F}}_{L,\kappa}^+(\gamma,\nu) \subset \mathcal{F}_{L,\kappa}$. Thereby, by applying Talagrand's lemma, (14) leads to

$$\mathbb{E}\left[ \ell_{2,\gamma}(\hat{f}_k(\boldsymbol{X})) \mid Y = k \right] \leq \frac{1}{n_k} \sum_{y_i=k} \ell_{2,\gamma}(\hat{f}_k(\boldsymbol{x}_i)) + 2 \cdot \mathfrak{R}_{n_k}(\pi_k \circ \mathcal{F}_{L,\kappa}) + r\sqrt{\frac{2}{n_k} \log \frac{2K}{\delta}}$$

$$:= \frac{1}{n_k} \sum_{y_i=k} \ell_{2,\gamma}(\hat{f}_k(\boldsymbol{x}_i)) + \vartheta_{n_k}(\delta), \tag{15}$$

where

$$\mathfrak{R}_{n_k}(\pi_k \circ \mathcal{F}_{L,\kappa}) = \frac{4}{\sqrt{n_k}} + \frac{12r \log(r\sqrt{n_k})}{\sqrt{n_k}} \cdot \left[ \sum_{l=1}^{L} (m_l \ln(2m_l p_l))^{\frac{1}{3}} \right]^{\frac{3}{2}} \tag{16}$$

by applying Theorem 4.

**Proof of Theorem 3**: We skip the proof of Statement (1) since it can be immediately derived from Eq. (15) with the hypothesis class $\widehat{\mathcal{F}}_{L,\kappa}^+(\gamma - \nu - \vartheta^*, \nu)$. Now let's work on the proof of Statement (2),

which will be divided into 2 parts. We define $\hat{\boldsymbol{f}}, \bar{\boldsymbol{f}} \in \widehat{\mathcal{F}}_{L,\boldsymbol{\kappa}}^+(\gamma - \nu - \vartheta^*, \nu)$ be the minimizer of $\widehat{\mathcal{R}}_{\ell_1}(\boldsymbol{f}) := \frac{1}{n} \sum_{i=1}^n \sum_{k=1}^K \ell_1(f_k(\boldsymbol{x}_i))$ and $\frac{1}{K} \sum_{k=1}^K \mathbb{E}[\ell_1(f_k(\boldsymbol{X}))]$, respectively. WLOG, we assume the loss function $\ell_1$ has the Lipschitz constant $c$. Note that the operation $\sum_{k=1}^K \ell_1(f_k(\boldsymbol{x}_i))$ here can be viewed as $\mathbf{1}^\top \circ \ell_1 \circ \boldsymbol{f}(\boldsymbol{x}_i)$, where $\mathbf{1}$ is the vector with all $K$ entries taking the value 1; $\ell_1$ is element-wisely applied on $\boldsymbol{f}(\boldsymbol{x}_i)$ and the composite mapping $\mathbf{1}^\top \circ \ell_1$ has Lipschitz constant $\sqrt{K}c$ due to the fact that $\forall \boldsymbol{z} = (z_1, \cdots, z_K)^\top$ and $\tilde{\boldsymbol{z}} = (\tilde{z}_1, \cdots, \tilde{z}_K)^\top$,

$$\|\mathbf{1}^\top \circ \ell_1(\boldsymbol{z}) - \mathbf{1}^\top \circ \ell_1(\tilde{\boldsymbol{z}})\|_2 \leq \|\mathbf{1}^\top\|_2 \cdot \|\ell_1(\boldsymbol{z}) - \ell_1(\tilde{\boldsymbol{z}})\|_2$$
$$= \sqrt{K} \cdot \sqrt{\sum_{k=1}^K \big(\ell_1(z_k) - \ell_1(\tilde{z}_k)\big)^2} \leq \sqrt{K}c \sqrt{\sum_{k=1}^K (z_k - \tilde{z}_k)^2} = \sqrt{K}c \cdot \|\boldsymbol{z} - \tilde{\boldsymbol{z}}\|_2.$$

i) This part is to bound $\mathcal{R}_{\ell_1}(\hat{\boldsymbol{f}}) - \min\limits_{\boldsymbol{f} \in \mathcal{F}_{L,\boldsymbol{\kappa}}^+(\gamma - \tilde{\nu} - 2\vartheta^*, \tilde{\nu})} \mathcal{R}_{\ell_1}(\boldsymbol{f})$, where $\tilde{\nu} \in [0, \gamma - 2\vartheta^*]$.

Let's first prove the statement: with probability at least $1 - \delta$,

$$\mathcal{F}_{L,\boldsymbol{\kappa}}^+(\gamma - \tilde{\nu} - 2\vartheta^*, \tilde{\nu}) \subset \widehat{\mathcal{F}}_{L,\boldsymbol{\kappa}}^+(\gamma - \nu - \vartheta^*, \nu). \tag{17}$$

*Proof.* On the one hand, for any $\tilde{\boldsymbol{f}} \in \mathcal{F}_{L,\boldsymbol{\kappa}}^+(\gamma - \tilde{\nu} - 2\vartheta^*, \tilde{\nu})$, we have

$$\mathbb{P}(\tilde{f}_k(\boldsymbol{X}) < 0 \mid Y = k) - (\gamma - \tilde{\nu} - 2\vartheta^*) \leq \mathbb{E}[\ell_{\gamma - \tilde{\nu} - 2\vartheta^*}(\tilde{f}_k(\boldsymbol{X})) \mid Y = k] \leq \tilde{\nu}$$
$$\Rightarrow \mathbb{P}(\tilde{f}_k(\boldsymbol{X}) < 0 \mid Y = k) \leq \gamma - 2\vartheta^*.$$

On the other hand, for any $\breve{\boldsymbol{f}} \in \widehat{\mathcal{F}}_{L,\boldsymbol{\kappa}}^+(\gamma - \nu - \vartheta^*, \nu)$, similar to the prove of Eq. (15), with low probability at most $\delta$, we have

$$\mathbb{E}[\ell_{\gamma - \nu - \vartheta^*}(\breve{f}_k(\boldsymbol{X})) \mid Y = k] \leq \frac{1}{n_k} \sum_{y_i = k} \ell_{\gamma - \nu - \vartheta^*}(\breve{f}_k(\boldsymbol{x}_i)) - \vartheta_{n_k}(\delta)$$
$$\leq \nu - \vartheta_{n_k}(\delta)$$
$$\Rightarrow \mathbb{P}(\breve{f}_k(\boldsymbol{X}) < 0 \mid Y = k) - (\gamma - \nu - \vartheta^*) \leq \nu - \vartheta_{n_k}(\delta)$$
$$\Rightarrow \mathbb{P}(\breve{f}_k(\boldsymbol{X}) < 0 \mid Y = k) \leq \gamma - \vartheta^* - \vartheta_{n_k}(\delta),$$

thus with probability at least $1 - \delta$, we have

$$\mathbb{P}(\breve{f}_k(\boldsymbol{X}) < 0 \mid Y = k) > \gamma - \vartheta^* - \vartheta_{n_k}(\delta) \geq \gamma - 2\vartheta^* \geq \mathbb{P}(\tilde{f}_k(\boldsymbol{X}) < 0 \mid Y = k),$$

which implies $\mathcal{F}_{L,\boldsymbol{\kappa}}^+(\gamma - \tilde{\nu} - 2\vartheta^*, \tilde{\nu}) \subset \widehat{\mathcal{F}}_{L,\boldsymbol{\kappa}}^+(\gamma - \nu - \vartheta^*, \nu)$.

Therefore, with probability $1 - \delta$, the following inequality holds

$$\mathcal{R}_{\ell_1}(\hat{\boldsymbol{f}}) - \min_{\boldsymbol{f} \in \mathcal{F}_{L,\boldsymbol{\kappa}}^+(\gamma - \tilde{\nu} - 2\vartheta^*, \tilde{\nu})} \mathcal{R}_{\ell_1}(\boldsymbol{f}) = \mathcal{R}_{\ell_1}(\hat{\boldsymbol{f}}) - \widehat{\mathcal{R}}_{\ell_1}(\hat{\boldsymbol{f}}) + \widehat{\mathcal{R}}_{\ell_1}(\bar{\boldsymbol{f}}) - \mathcal{R}_{\ell_1}(\bar{\boldsymbol{f}})$$
$$+ \widehat{\mathcal{R}}_{\ell_1}(\hat{\boldsymbol{f}}) - \widehat{\mathcal{R}}_{\ell_1}(\bar{\boldsymbol{f}}) \tag{18}$$
$$+ \mathcal{R}_{\ell_1}(\bar{\boldsymbol{f}}) - \min_{\boldsymbol{f} \in \mathcal{F}_{L,\boldsymbol{\kappa}}^+(\gamma - \tilde{\nu} - 2\vartheta^*, \tilde{\nu})} \mathcal{R}_{\ell_1}(\boldsymbol{f}) \tag{19}$$
$$\leq 2 \sup_{\boldsymbol{f} \in \mathcal{F}_{L,\boldsymbol{\kappa}}} |\mathcal{R}_{\ell_1}(\boldsymbol{f}) - \widehat{\mathcal{R}}_{\ell_1}(\boldsymbol{f})|, \tag{20}$$

because Eq. (18) is non-positive ($\hat{\boldsymbol{f}}$ is an empirical minimizer), and Eq. (19) is non-positive with probability $1 - \delta$ due to the fact of Eq. (17) and $\bar{\boldsymbol{f}}$ is a minimizer in $\widehat{\mathcal{F}}_{L,\boldsymbol{\kappa}}^+(\gamma - \nu - \vartheta^*, \nu)$.

Let $\mathcal{F}_{L,\boldsymbol{\kappa}}^{sum,\ell_1} := \{\boldsymbol{x} \mapsto \sum_{k=1}^{K} \ell_1(f_k(\boldsymbol{x})) : \boldsymbol{f} = (f_1, \cdots, f_K) \in \mathcal{F}_{L,\boldsymbol{\kappa}}\}$ be a space in which a Lipschitz continuous function (the Lipschitz constant is $\sqrt{K}c$) is applied on the function vector $\boldsymbol{f} \in \mathcal{F}_{L,\boldsymbol{\kappa}}$. Thus, by applying Theorem 4, we have

$$\mathfrak{R}_n(\mathcal{F}_{L,\boldsymbol{\kappa}}^{sum,\ell_1}) \leq \frac{4}{\sqrt{n}} + \frac{12\sqrt{K}c \cdot r \log(r\sqrt{n})}{\sqrt{n}} \cdot \left[\sum_{l=1}^{L} (m_l \ln(2m_l p_l))^{\frac{1}{3}}\right]^{\frac{3}{2}}, \tag{21}$$

where $c = 1$ when we particularly work with the hinge loss.

Following the similar proof for Eq. (14) (now the bounded difference is $2\sqrt{K}cr/n$), with probability $1 - \delta$ we have

$$\sup_{\boldsymbol{f} \in \mathcal{F}_{L,\boldsymbol{\kappa}}} |\mathcal{R}_{\ell_1}(\boldsymbol{f}) - \widehat{\mathcal{R}}_{\ell_1}(\boldsymbol{f})| \leq 2\mathfrak{R}_n(\mathcal{F}_{L,\boldsymbol{\kappa}}^{sum,\ell_1}) + 2\sqrt{K}cr\sqrt{\frac{1}{2n}\log\frac{2}{\delta}} \tag{22}$$

and hence with probability at least $1 - 2\delta$, (20) and (22) yields

$$\mathcal{R}_{\ell_1}(\hat{\boldsymbol{f}}) - \min_{\boldsymbol{f} \in \mathcal{F}_{L,\boldsymbol{\kappa}}^+(\gamma-\tilde{\nu}-2\vartheta^*,\tilde{\nu})} \mathcal{R}_{\ell_1}(\boldsymbol{f}) \leq 4\mathfrak{R}_n(\mathcal{F}_{L,\boldsymbol{\kappa}}^{sum,\ell_1}) + 4\sqrt{K}cr\sqrt{\frac{1}{2n}\log\frac{2}{\delta}}. \tag{23}$$

ii) This part is to bound $\min_{\boldsymbol{f} \in \mathcal{F}_{L,\boldsymbol{\kappa}}^+(\gamma-\tilde{\nu}-2\vartheta^*,\tilde{\nu})} \mathcal{R}_{\ell_1}(\boldsymbol{f}) - \min_{\boldsymbol{f} \in \mathcal{F}_{L,\boldsymbol{\kappa}}^+(\gamma-\tilde{\nu},\tilde{\nu})} \mathcal{R}_{\ell_1}(\boldsymbol{f})$.

Let $\boldsymbol{f}^{in} = \operatorname*{argmin}_{\boldsymbol{f} \in \mathcal{F}_{L,\boldsymbol{\kappa}}^+(\gamma-\tilde{\nu}-2\vartheta^*,\tilde{\nu})} \mathcal{R}_{\ell_1}(\boldsymbol{f})$ and $\boldsymbol{f}^{out} = \operatorname*{argmin}_{\boldsymbol{f} \in \mathcal{F}_{L,\boldsymbol{\kappa}}^+(\gamma-\tilde{\nu},\tilde{\nu})} \mathcal{R}_{\ell_1}(\boldsymbol{f})$. Note that $\boldsymbol{f}^{in}, \boldsymbol{f}^{out} \in \mathcal{F}_{L,\boldsymbol{\kappa}}$. Thus

$$\begin{aligned}
&\min_{\boldsymbol{f} \in \mathcal{F}_{L,\boldsymbol{\kappa}}^+(\gamma-\tilde{\nu}-2\vartheta^*,\tilde{\nu})} \mathcal{R}_{\ell_1}(\boldsymbol{f}) - \min_{\boldsymbol{f} \in \mathcal{F}_{L,\boldsymbol{\kappa}}^+(\gamma-\tilde{\nu},\tilde{\nu})} \mathcal{R}_{\ell_1}(\boldsymbol{f}) \\
&= \mathcal{R}_{\ell_1}(\boldsymbol{f}^{in}) - \mathcal{R}_{\ell_1}(\boldsymbol{f}^{out}) \\
&= \mathcal{R}_{\ell_1}(\boldsymbol{f}^{in}) - \widehat{\mathcal{R}}_{\ell_1}(\boldsymbol{f}^{in}) + \widehat{\mathcal{R}}_{\ell_1}(\boldsymbol{f}^{out}) - \mathcal{R}_{\ell_1}(\boldsymbol{f}^{out}) + \widehat{\mathcal{R}}_{\ell_1}(\boldsymbol{f}^{in}) - \widehat{\mathcal{R}}_{\ell_1}(\boldsymbol{f}^{out}) \\
&\leq 2 \sup_{\boldsymbol{f} \in \mathcal{F}_{L,\boldsymbol{\kappa}}} |\mathcal{R}_{\ell_1}(\boldsymbol{f}) - \widehat{\mathcal{R}}_{\ell_1}(\boldsymbol{f})| + \widehat{\mathcal{R}}_{\ell_1}(\boldsymbol{f}^{in}) - \widehat{\mathcal{R}}_{\ell_1}(\boldsymbol{f}^{out})
\end{aligned} \tag{24}$$

Define $\widetilde{\mathcal{R}}(\boldsymbol{f}^{in}, \boldsymbol{f}^{out}) := \widehat{\mathcal{R}}_{\ell_1}(\boldsymbol{f}^{in}) - \widehat{\mathcal{R}}_{\ell_1}(\boldsymbol{f}^{out}) = \frac{1}{n}\sum_{i=1}^{n}\sum_{k=1}^{K} \ell_l(f_k^{in}(\boldsymbol{x}_i)) - \ell_1(f_k^{out}(\boldsymbol{x}_i))$. Since

$$\begin{aligned}
&\left| \sum_{k=1}^{K} \ell_l(f_k^{in}(\boldsymbol{x}_i)) - \ell_1(f_k^{out}(\boldsymbol{x}_i)) - \sum_{k=1}^{K} \ell_l(f_k^{in}(\boldsymbol{x}_i')) - \ell_1(f_k^{out}(\boldsymbol{x}_i')) \right| \\
&= \left| \sum_{k=1}^{K} \ell_l(f_k^{in}(\boldsymbol{x}_i)) - \ell_1(f_k^{in}(\boldsymbol{x}_i')) - \sum_{k=1}^{K} \ell_l(f_k^{out}(\boldsymbol{x}_i)) - \ell_1(f_k^{out}(\boldsymbol{x}_i')) \right| \\
&\leq \sum_{k=1}^{K} \left| \ell_l(f_k^{in}(\boldsymbol{x}_i)) - \ell_1(f_k^{in}(\boldsymbol{x}_i')) \right| + \sum_{k=1}^{K} \left| \ell_l(f_k^{out}(\boldsymbol{x}_i)) - \ell_1(f_k^{out}(\boldsymbol{x}_i')) \right| \\
&\leq 2\sum_{k=1}^{K} \left| \ell_l(f_k(\boldsymbol{x}_i)) - \ell_1(f_k(\boldsymbol{x}_i')) \right| \qquad \text{here } \boldsymbol{f} = (f_1, \ldots, f_K)^\top \in \mathcal{F}_{L,\boldsymbol{\kappa}} \\
&\leq 2\sum_{k=1}^{K} c|f_k(\boldsymbol{x}_i) - f_k(\boldsymbol{x}_i')| \qquad\qquad \ell_1 \text{ is a } c\text{-Lipschitz continuous function} \\
&\leq 2c\sum_{k=1}^{K} \|(\mathbf{W}_L)_{k,\cdot}\|_2 \cdot \left(\prod_{l=1}^{L-1} \kappa_l\right) \|\boldsymbol{x}_i - \boldsymbol{x}_i'\|_2 \qquad (\mathbf{W}_L)_{k,\cdot} \text{ denotes } k\text{-th row of matrix } \mathbf{W}_L \\
&\leq 4c\sqrt{K}c_0 \prod_{l=1}^{L} \kappa_l = 4c\sqrt{K}cr,
\end{aligned}$$

by McDiarmid's inequality (the bounded difference is $4\sqrt{K}cr/n$), with the probability at least $1-\delta$, we have

$$
\begin{aligned}
\widetilde{\mathcal{R}}(\boldsymbol{f}^{in}, \boldsymbol{f}^{out}) &\leq \mathbb{E}[\widetilde{\mathcal{R}}(\boldsymbol{f}^{in}, \boldsymbol{f}^{out})] + 4\sqrt{K}cr\sqrt{\frac{1}{2n}\log\frac{2}{\delta}} \\
&\leq 2\mathfrak{R}_n\left(\boldsymbol{x} \mapsto \sum_{k=1}^{K} \ell_1(f_k^{in}(\boldsymbol{x})) - \ell_1(f_k^{out}(\boldsymbol{x}))\right) + 4\sqrt{K}cr\sqrt{\frac{1}{2n}\log\frac{2}{\delta}} \\
&\leq 4\mathfrak{R}_n(\mathcal{F}_{L,\boldsymbol{\kappa}}^{sum,\ell_1}) + 4\sqrt{K}cr\sqrt{\frac{1}{2n}\log\frac{2}{\delta}}
\end{aligned}
\tag{25}
$$

Therefore, together with (22) and (25), (24) can be bounded as follows:

$$
\begin{aligned}
&\min_{\boldsymbol{f}\in\mathcal{F}_{L,\boldsymbol{\kappa}}^+(\gamma-\tilde{\nu}-2\vartheta^*,\tilde{\nu})} \mathcal{R}_{\ell_1}(\boldsymbol{f}) - \min_{\boldsymbol{f}\in\mathcal{F}_{L,\boldsymbol{\kappa}}^+(\gamma-\tilde{\nu},\tilde{\nu})} \mathcal{R}_{\ell_1}(\boldsymbol{f}) \\
&\leq 2\sup_{\boldsymbol{f}\in\mathcal{F}_{L,\boldsymbol{\kappa}}} |\mathcal{R}_{\ell_1}(\boldsymbol{f}) - \widehat{\mathcal{R}}_{\ell_1}(\boldsymbol{f})| + \widehat{\mathcal{R}}_{\ell_1}(\boldsymbol{f}^{in}) - \widehat{\mathcal{R}}_{\ell_1}(\boldsymbol{f}^{out}) \\
&\leq 4\mathfrak{R}_n(\mathcal{F}_{L,\boldsymbol{\kappa}}^{sum,\ell_1}) + 4\sqrt{K}cr\sqrt{\frac{1}{2n}\log\frac{2}{\delta}} + 4\mathfrak{R}_n(\mathcal{F}_{L,\boldsymbol{\kappa}}^{sum,\ell_1}) + 4\sqrt{K}cr\sqrt{\frac{1}{2n}\log\frac{2}{\delta}} \\
&= 8\mathfrak{R}_n(\mathcal{F}_{L,\boldsymbol{\kappa}}^{sum,\ell_1}) + 8\sqrt{K}cr\sqrt{\frac{1}{2n}\log\frac{2}{\delta}}.
\end{aligned}
\tag{26}
$$

Combining (23) in Part (i) and (26) in Part (ii), we conclude that, with probability at least $1-3\delta$,

$$
\begin{aligned}
&\mathcal{R}_{\ell_1}(\hat{\boldsymbol{f}}) - \min_{\boldsymbol{f}\in\mathcal{F}_{L,\boldsymbol{\kappa}}^+(\gamma-\tilde{\nu},\tilde{\nu})} \mathcal{R}_{\ell_1}(\boldsymbol{f}) \\
&\leq 4\mathfrak{R}_n(\mathcal{F}_{L,\boldsymbol{\kappa}}^{sum,\ell_1}) + 4\sqrt{2}cr\sqrt{\frac{1}{2n}\log\frac{K}{\delta}} + 8\mathfrak{R}_n(\mathcal{F}_{L,\boldsymbol{\kappa}}^{sum,\ell_1}) + 8\sqrt{K}cr\sqrt{\frac{1}{2n}\log\frac{2}{\delta}} \\
&\leq 12\mathfrak{R}_n(\mathcal{F}_{L,\boldsymbol{\kappa}}^{sum,\ell_1}) + 12\sqrt{K}cr\sqrt{\frac{1}{2n}\log\frac{2}{\delta}},
\end{aligned}
\tag{27}
$$

where $c = 1$ if we particularly use the hinge loss. $\qquad\square$

