# OpenReview forum: "Deep Generalized Prediction Set Classifier and Its Theoretical Guarantees"
_TMLR — Accepted by TMLR_

### Review · Reviewer_2jdX · 2024-02-24

**Summary Of Contributions:**

The paper proposes a generalized prediction set classifier, which returns set valued classifier to handle the uncertainty in classification. The proposed method uses a fully connected neural network as the hypothesis function, and uses random features to approximate kernel mapping. The proposed objective function consists of three terms reflecting the set size, the accuracy and the regularity. The paper presents theoretical results to illustrate the behavior of the proposed algorithm. The first theoretical result is the approximation error on approximating the kernel by random feature. The second result is the generalization bounds.

Experimental results with several baselines and practical datasets are reported to show the empirical behavior of the proposed method.

**Audience:**

Yes

**Broader Impact Concerns:**

No concerns on the ethical issues.

**Claims And Evidence:**

Yes

**Requested Changes:**

- The authors should give details of the theoretical analysis in a step-by-step way. This would make be improve the readability of the paper.
- Notations should be clearly defined and the explicit reference should be given.

**Strengths And Weaknesses:**

**Strength**

- The set-valued prediction is interesting to handle uncertainty and OOD detection. The problem considered is interesting, general and there is not enough study for this problem in the literature.
- The paper includes both theoretical analysis and experimental analysis. The paper includes extensive experimental results and experimental comparisons. The experimental results seem to be strong and convincing.

**Weakness**
- In the proof of Theorem 3, the paper uses the inequality $R_n(F_{L,\kappa}^{sum})\leq \sqrt{Kc}R_n(F_{L,\kappa})$. It is not clear to me how this holds. Anyhow, the definition of $R_n(F_{L,\kappa})$ is not given. Furthermore, as far as I see, F_{L,\kappa} is a vector-valued function class. Therefore, the authors should give the explicit meaning of the Rademacher complexity.
- The theoretical analysis ignores several steps. For example, in the proof of Theorem 1, the paper directly uses the inequality $R_D(\xi\mapsto \sigma\xi^\top\triangle)\leq \prod_l\kappa_l/\sqrt{D}$. It is not clear to me how this holds.
- Several notations are not well-defined. For example, the covering number in Lemma 1 is not defined. The Rademacher complexity is also not defined.
- Some results are used without giving the reference, e.g., Lemma 1 and Lemma 2.

---

> ### Author Response · Authors · 2024-03-26
>
> > ...... It is not clear to me how this holds ...... F_{L,\kappa} is a vector-valued function class. Therefore, the authors should give the explicit meaning of the Rademacher complexity
>
> We apologize for having skipped a few steps in the proof. We sincerely thank you for your critical and valuable input.  We have defined the Rademacher complexity and the covering number for $\mathcal{F}_{L, \kappa}$ and corrected the proofs in our revision (see Appendix D).
>
> You are right that we cannot transfer the properties of the Rademacher complexity of the real-valued function class to the vector-valued function class. We have revised our proof in the revision (see the appendix). TL;DR: Our new proof no longer uses the Rademacher complexity of the vector-valued function class. Instead, we first apply the Lipschitz mapping to the covering number of the vector-valued function class (the notion of the covering number can extend to the vector-valued function class) to obtain the covering number of the real-valued function class. Then, by applying Dudley’s Theorem, we can obtain the Rademacher complexity of the real-valued function class. Additionally, we have provided more details when utilizing some properties of Rademacher complexity, e.g., Talagrand’s lemma.
>
> > The theoretical analysis ignores several steps...... paper directly uses the inequality $R_D(\xi\mapsto\sigma\xi^\top\Delta)\leq \prod_l\kappa_l \sqrt{D}$......
>
> We apologize for the undetailed proof. We have provided more details to our proofs in the revision (see the new Lemma 1). Briefly speaking, we rely on the Rademacher complexity for $\ell_2$ ball of $\Delta$ where $\Delta$ is the difference between the representations of $\boldsymbol{x}_i$ and $\boldsymbol{x}_j$ in the $(L-2)$-th hidden layer of the neural network. This Rademacher complexity is bounded by $\sigma\sqrt{\mathbb{E}[\|\boldsymbol{\xi}\|_2^2]/D}$  times $\|\Delta\|_2$. To bound the $\|\Delta\|_2$, we need a Lipschitz-type property of the network (which is a composition of Lipschitz mappings, including activation functions and matrix operations), where the Lipschitz constants of activation functions are all 1 and the Lipschitz constants of matrix operations are characterized by $\kappa_l$.
>
> > Several notations are not well-defined. For example, the covering number in Lemma 1 is not defined. The Rademacher complexity is also not defined
>
>  We appreciate the reviewer's suggestion. We have added the definitions for both covering number and the Rademacher complexity, along with several other notations, to make the paper more self-contained.
>
>
> > Some results are used without giving the reference, e.g., Lemma 1 and Lemma 2.
>
>  We apologize for this oversight. We have added the reference in the revision.
>
> > The authors should give details of the theoretical analysis in a step-by-step way. This would make be improve the readability of the paper.
>
>   We appreciate the reviewer's helpful suggestion. We have refined our steps of proof in our revision, including those mentioned in weaknesses \#1 and \#2.
>
> > Notations should be clearly defined and the explicit reference should be given.
>
> We thank the reviewer for constructive comments. We have provided detailed definitions to notations and added the related literature in the proofs, including those mentioned above in weaknesses \#3 and \#4.

---

> > ### Comment · Reviewer_2jdX · 2024-03-30
> > **Thank you for your revision**
> >
> > I thank the authors for incorporating all my previous suggestions into their revision of the paper. The paper contains both comprehensive theoretical and empirical analysis. I recommend the acceptance of paper for publication in TMLR.

---

### Review · Reviewer_Bzm6 · 2024-03-19

**Summary Of Contributions:**

The paper presents the Deep Generalized Prediction Set (DeepGPS) classifier, a deep network-based set-valued classifier for handling uncertainty in classification by identifying ambiguous observations and detecting out-of-distribution (OOD) observations. DeepGPS employs three proposed terms (two loss functions $\ell_1$, $\ell_{2, \gamma}$ and a regularization $J$ in Equation (1)) as an objective to minimize the ambiguity in the observation, both in the labeled data (classes 1 to K) and the unlabeled data (OOD data). There are three theoretical results: the convergence of the kernel approximation (Theorem 1), the generalization error bound of the class-wise loss (measured by $\ell_{2, \gamma}$, classes 1 to K), and the excess risk measured by $\ell_1$ (including OOD).

**Audience:**

Yes

**Claims And Evidence:**

No

**Requested Changes:**

Questions:

Q1: How did the authors find the value for the hyper-parameter $q$ in experiments?

Q2: Can the authors analyze the Assumption 1 and 2 for comparing them with other literatures in OOD? Assumption 1 and 2 together look a bit strong.

Q3: The paper mentions OOD detection as a feature of DeepGPS. Can the authors elaborate in detail on how DeepGPS can be generalized to other potential capabilities?

**Strengths And Weaknesses:**

Strengths:

This paper proposes an original objective function with theoretical analysis for the set-predictor classifier, where the ambiguity is measured and minimized on the labeled data and unlabeled data, particularly for OOD problems. The paper is clear and easy to follow. The proposed techniques are mostly well motivated and addressed. Approximation and error bounds are well handled, in the standard order of $\tilde O(1/\sqrt{n})$.

Weaknesses:

1. Experiments include only some small-scale datasets. More realistic experiments would be highly recommended.
2. Theorem 1 looks separate from the remaining Theorem 2 and Theorem 3, and also may not be connected well to the entire design. For example, how Theorem 1 benefits the generalization error bound or excess risk? How Theorem 1 helps the experiment performance?
3. Figure 2 and Figure 3 are important conceptual figures to support the design, but both figures are too informative to easily connect to the main text. More explanation and captions can be very helpful. For example, in Figure 2, linking the weight function and weighted loss to reference equations. What is the value of $\gamma$ in Figure 2?

---

> ### Author Response · Authors · 2024-03-26
> **Part 1**
>
> > Experiments include only some small-scale datasets. More realistic experiments would be highly recommended.
>
> We sincerely value the feedback regarding our experimental datasets. Note that our work aligns with existing literature in set-valued classification and OOD detection, which predominantly employs similar or even simpler datasets, e.g. non-image data. Acknowledging the merit of the suggestion, we have incorporated the experiment on another large data CIFAR-100 in Table 2 (see Appendix C). It shows that the proposed DeepGPS has a better trade-off between OOD recall and efficiency, in addition to the desired control of class-specific accuracy.
>
> > Theorem 1 looks separate from the remaining Theorem 2 and Theorem 3, and also may not be connected well to the entire design. For example, how Theorem 1 benefits the generalization error bound or excess risk? How Theorem 1 helps the experiment performance?
>
> We thank the reviewer's valuable questions. Theorem 1 does not theoretically affect Theorems 2 and 3. The network studied by Theorem 1 is just a special case of those more generic networks studied in Theorems 2 and 3 (in particular, the second last hidden layer is the kernel approximation layer, and the activation function is sinusoidal instead of ReLU). Our proof for Theorem 2 and 3 only relies on the norm of weight matrices and the Lipschitzness of activation functions in the hidden layers.
>
> While Theorem 1 might appear distinct, its role in underpinning the geometric intuition---behind our design of the kernel approximation layer---is crucial for understanding the empirical effectiveness of DeepGPS. Specifically, by observing the geometric interpretation of the Gaussian kernel in OCSVM as in the discussion of Figure 4, a kernel machine often returns a closed and compact acceptance region in practice (see Figure 7 for an example), which leads to smaller prediction sets and improves the OOD detection performance.
>
> > Figure 2 and Figure 3 are important conceptual figures to support the design, but both figures are too informative to easily connect to the main text. More explanation and captions can be very helpful. For example, in Figure 2, linking the weight function and weighted loss to reference equations. What is the value of
> $\gamma$ in Figure 2?
>
> We greatly appreciate the reviewer's constructive suggestions to make our work more engaging. We have revised Figures 2 and 3 to make them more accessible and relatable to the main text by enriching their captions and directly connecting them to relevant equations and concepts in the discussion. The value of $\gamma=0.1$ in Figure 2 is set for illustration.
>
> > How did the authors find the value for the hyper-parameter $q$ in experiments?
>
>
> We are grateful for your comment concerning the selection of the hyper-parameter $q$. In our experiments, we opted for a uniform value of $q=5$ across all datasets, without individual tuning to maintain methodological consistency. To show its impact, we added a sensitivity analysis in Figure 8 and Page 11, including the case not using weighted loss, along with $q=5$ and 10. The results between $q=5$ and 10 are very close, and both are significantly different from the case with unweighted loss. Hence in practice, we recommend fixing the value of $q$ to be a reasonable positive value such as 5.

---

> ### Author Response · Authors · 2024-03-26
> **Part 2**
>
> > Can the authors analyze the Assumption 1 and 2 for comparing them with other literatures in OOD? Assumption 1 and 2 together look a bit strong.
>
> We thank the reviewers' thoughtful comments on our assumptions. Assumption 1 aligns with established assumptions in the OOD detection and open-set recognition literature [1, 2, 3], while Assumption 2 is a similarly foundational work addressing new classes in the test data [3, 4, 5].
>
> We acknowledge that these assumptions, especially in conjunction, may appear stringent. To relax these assumptions, we have investigated future explorations that may adopt alternative assumptions more reflective of varied real-world conditions.
>
> * As per the relaxing of Assumption 1 of equal class-conditional densities, we can follow the work in [6, 7] by assuming there is a location and scale shift of covariates given a class across the training and test data. The kernel mean embedding matching approach [8, 9] allows us to tractably handle this type of different class-conditional distributions.
> * As per the relaxing of Assumption 2 of the additional unlabeled data, DeepGPS still works with the generated auxiliary data [10, 11] around labeled normal observations even if one has no access to the test data.
>      On the other hand, without accessing all data points, DeepGPS's application extends seamlessly to online learning environments, where the model is updated by optimizing objective function (1) with stochastic gradient descent on each batch of data. Under the relaxing of Assumption 2, the exploration of DeepGPS's performance in both synthetic data and its efficacy within online learning setups would be future works to show its adaptability.
>
>  We thank you for this valuable point since it offers us a promising future work direction. We have added detailed discussions and the possible strategies to relax Assumptions 1 and 2 in Remark 1 in the revision.
>
>  [1] Yang, Jingkang, et al. "Generalized out-of-distribution detection: A survey." arXiv preprint arXiv:2110.11334 (2021).
>
>  [2] Garg, Saurabh, Sivaraman Balakrishnan, and Zachary Lipton. "Domain adaptation under open set label shift." Advances in Neural Information Processing Systems 35 (2022): 22531-22546.
>
>  [3] Katz-Samuels, Julian, et al. "Training ood detectors in their natural habitats." International Conference on Machine Learning. PMLR, 2022.
>
>  [4] Du Plessis, Marthinus, Gang Niu, and Masashi Sugiyama. "Convex formulation for learning from positive and unlabeled data." International conference on machine learning. PMLR, 2015.
>
>  [5] Guan, Leying, and Robert Tibshirani. "Prediction and outlier detection in classification problems." Journal of the Royal Statistical Society Series B: Statistical Methodology 84.2 (2022): 524-546.
>
>  [6] Zhang, Kun, et al. "Domain adaptation under target and conditional shift." International conference on machine learning. Pmlr, 2013.
>
>  [7] Gong, Mingming, et al. "Domain adaptation with conditional transferable components." International conference on machine learning. PMLR, 2016.
>
>  [8] Fukumizu, Kenji, et al. "Kernel measures of conditional dependence." Advances in neural information processing systems 20 (2007).
>
>  [9] Sriperumbudur, Bharath K., Kenji Fukumizu, and Gert RG Lanckriet. "Universality, Characteristic Kernels and RKHS Embedding of Measures." Journal of Machine Learning Research 12.7 (2011).
>
> [10]  D. Hendrycks, M. Mazeika, and T. G. Dietterich, “Deep anomaly detection with outlier exposure,” In ICLR, 2019.
>
> [11] Xuefeng Du, Zhaoning Wang, Mu Cai, and Yixuan Li. Vos: Learning what you don’t know by virtual
> outlier synthesis. In ICLR, 2022.
>
> > The paper mentions OOD detection as a feature of DeepGPS. Can the authors elaborate in detail on how DeepGPS can be generalized to other potential capabilities?
>
> We are indeed grateful for the opportunity to discuss the versatility and potential expansions of DeepGPS beyond its foundational application in Out-of-Distribution (OOD) detection.
>
> Firstly, as discussed previously, DeepGPS together with the kernel mean matching mechanism addresses generalized label shift issues in domain adaptation tasks,  where class-conditional densities differ between training and test distributions.
>
>
> Secondly, DeepGPS could facilitate active learning processes by identifying instances where the model is uncertain (i.e., cases that might be near the decision boundary or completely OOD). These instances can then be prioritized for labeling by human experts, efficiently utilizing resources and improving model performance with a constrained dataset.
>
> Lastly, integrating DeepGPS within chatbot systems could revolutionize personalized recommendation strategies, tailoring responses to align with user preferences and profiles. In instances of encountering unseen queries, the chatbot, empowered by DeepGPS, could judiciously opt for an "unknown/out-of-scope" response, thereby circumventing the delivery of irrelevant or inaccurate information.

---

> > ### Comment · Reviewer_Bzm6 · 2024-04-22
> > **Thanks for addressing my review comments**
> >
> > I would like to thank the authors for addressing my comments and questions. I'm happy to recommend acceptance of the paper.

---

### Review · Reviewer_XRkW · 2024-03-20

**Summary Of Contributions:**

This article presents Deep Generalized Prediction Set (DeepGPS), a novel network-based set-valued classifier that addresses the challenge of uncertainty in classification tasks. DeepGPS provides sets of plausible labels for ambiguous observations and effectively detects out-of-distribution (OOD) data points. Notably, DeepGPS is the first set-valued classification method to offer both theoretical guarantees and scalability for large datasets. The article provides a rigorous proof demonstrating DeepGPS's achievement of optimal risk within a neural network hypothesis class, while simultaneously ensuring user-prescribed class-specific accuracy. To improve its performance, DeepGPS incorporates a weighted loss mechanism, enabling the definition of tighter acceptance regions. Extensive empirical evaluations conducted on benchmark datasets confirm the superiority of DeepGPS over baseline methods.

**Audience:**

Yes

**Claims And Evidence:**

Yes

**Requested Changes:**

This paper stands out as a high-quality contribution, characterized by solid theoretical findings and substantial numerical experiments.I have only one minor comment: The authors may consider incorporating additional remarks regarding the utilization of alternative classification losses, such as cross entropy, logistic regression, and others.

**Strengths And Weaknesses:**

Strengths:

1. DeepGPS addresses the limitations of conventional single-valued predictions by providing confident set-valued decisions for interested classes.
2. The theoretical analysis demonstrates that DeepGPS minimizes the size of the prediction set while maintaining the prescribed accuracy.
3. DeepGPS incorporates OOD detection, making it effective in scenarios where accurate predictions are crucial to avoid severe consequences.
4. The offset penalization and weighted loss contribute to the construction of compact acceptance regions, improving the precision of predictions.


Weaknesses:

1. Although the hinge loss used in this paper have shown effectiveness, investigating other different classification loss functions may offer valuable insights and potentially enhance the performance of the proposed method.
2. In practice, Assumption 1 of this paper may be considered somewhat strong.

---

> ### Author Response · Authors · 2024-03-26
>
> > ...... investigating other different classification loss functions may offer valuable insights and potentially enhance the performance of the proposed method.
>
>  We sincerely value the reviewer's constructive suggestion regarding the exploration of alternative classification loss functions. Our initial selection of the hinge loss was motivated by its approximation to the 0-1 loss, aiming to efficiently minimize prediction set sizes without producing excessively large sets.
>
> To show the effectiveness of other loss functions, we have included the performance of DeepGPS on the CIFAR-10 with different surrogate losses (see Table 6 in Appendix C.4), i.e., logistic and exponential loss functions. Table 6 shows that the alternative two loss functions significantly sacrifice the prediction efficiency. This is mainly due to both logistic $\log(1+\exp(-u))$ and exponential loss $\exp(-u)$ having non-zero loss value when $u > 0$, which forces the prediction sets to include more labels such that the empirical loss values are below than the error bound $\gamma$ on average. In contrast, the hinge loss used in Table 1 returns smaller prediction sets without sacrificing too much OOD recall.
>
> Furthermore, we once considered the sigmoid function $\frac{1}{1+\exp(u)}$ as a candidate loss function which is closer to 0-1 loss than the hinge loss; however, the gradient vanishing with $u$ away from 0 hinders the training progress.
>
>
> > In practice, Assumption 1 of this paper may be considered somewhat strong.
>
> We thank the reviewer's valuable comment on this assumption. This assumption, about semantic shift, aligns with the premises of several seminal works in our field [1, 2, 3, 4]. That being said, your point has been definitely well-taken. We have explored relaxing Assumption 1 with other moderated assumptions, enabling a differential class-conditional density between the training and testing sets, which could enhance the model's adaptability to diverse domains.
>
> More concretely, the concepts of location-scale generalized target shift [5, 6] present a compelling framework for addressing domain adaptation challenges. By assuming an affine transformation across dimensions of $\boldsymbol X$ given $Y$, kernel embedding methods [7, 8] can be employed to align distributions between labeled training data and unlabeled test data. This offers a pathway for DeepGPS to reconcile distributional discrepancies.
>
> Encouraged by your feedback, we plan to further adapt DeepGPS to more complex and variable settings in future work. Detailed discussions on potential relaxations of Assumption 1 have been incorporated into Remark 1 of our revision.
>
>  [1] Du Plessis, Marthinus, Gang Niu, and Masashi Sugiyama. "Convex formulation for learning from positive and unlabeled data." International conference on machine learning. PMLR, 2015.
>
>  [2] Yang, Jingkang, et al. "Generalized out-of-distribution detection: A survey." arXiv preprint arXiv:2110.11334 (2021).
>
>  [3] Katz-Samuels, Julian, et al. "Training ood detectors in their natural habitats." International Conference on Machine Learning. PMLR, 2022.
>
>  [4] Garg, Saurabh, Sivaraman Balakrishnan, and Zachary Lipton. "Domain adaptation under open set label shift." Advances in Neural Information Processing Systems 35 (2022): 22531-22546.
>
> [5] Zhang, Kun, et al. "Domain adaptation under target and conditional shift." International conference on machine learning. Pmlr, 2013.
>
>  [6] Gong, Mingming, et al. "Domain adaptation with conditional transferable components." International conference on machine learning. PMLR, 2016.
>
>  [7] Fukumizu, Kenji, et al. "Kernel measures of conditional dependence." Advances in neural information processing systems 20 (2007).
>
>  [8] Sriperumbudur, Bharath K., Kenji Fukumizu, and Gert RG Lanckriet. "Universality, Characteristic Kernels and RKHS Embedding of Measures." Journal of Machine Learning Research 12.7 (2011).
>
> > The authors may consider incorporating additional remarks regarding the utilization of alternative classification losses, such as cross entropy, logistic regression, and others.
>
> Please see the response to the question in "Weakness 1".

---

### Decision · Action_Editor_37LP · 2024-05-01

**Recommendation:** Accept as is

**Comment:**

The paper proposed an end-to-end Deep Generalized Prediction Set (DeepGPS) classifier jointly learning acceptance regions with several contributions.  The first contribution is the it generalised the set-valued classification to OOD detection. The second one is that the authors  add the neural network a layer that approximates the kernel by using Random Fourier Features to overcome the computational overhead of the expensive memory and quadratic programming for kernel machines. The authors also provided learning theory guarantees and convincing experimental support for the proposed approach.

The revised version adequately addresses reviewers' questions/comments. All reviewers recommended acceptance and featured certification is recommended by one of the reviewers. I agree with the reviewers and recommend acceptance with featured certification.

**Audience:**

The problem of set-valued classification which can be generalised to out-of-distribution prediction is well motivated by application problems. It will be of great interest to TMLR audience.

**Claims And Evidence:**

Yes, the claims for the proposed new algorithm are supported by theory and experiments.

---

> ### Author Response · Authors · 2024-05-09
> **Thank you**
>
> Dear Action Editor,
>
> We want to express our gratitude to you and all reviewers for the dedicated time and thoughtful evaluations of our article. We truly appreciate all the constructive suggestions that have significantly enhanced the clarity, comprehensiveness, and soundness of our work.
>
> Following the provided instructions, we have uploaded the camera-ready version of our article. Thank you for your support and coordination throughout the review process.
>
> Sincerely,
>
> Authors